# The erosion of large primary atmospheres typically leaves behind substantial secondary atmospheres on temperate rocky planets

Joshua Krissansen-Totton [1,2] ✉, Nicholas Wogan [2,3], Maggie Thompson[4,5] & Jonathan J. Fortney[6]

Exoplanet exploration has revealed that many—perhaps most—terrestrial exoplanets formed with substantial $H_2$-rich envelopes, seemingly in contrast to solar system terrestrials, for which there is scant evidence of long-lived primary atmospheres. It is not known how a long-lived primary atmosphere might affect the subsequent habitability prospects of terrestrial exoplanets. Here, we present a new, self-consistent evolutionary model of the transition from primary to secondary atmospheres. The model incorporates all Fe-C-O-H-bearing species and simulates magma ocean solidification, radiative-convective climate, thermal escape, and mantle redox evolution. For our illustrative example TRAPPIST-1, our model strongly favors atmosphere retention for the habitable zone planet TRAPPIST-1e. In contrast, the same model predicts a comparatively thin atmosphere for the Venus-analog TRAPPIST-1b, which would be vulnerable to complete erosion via non-thermal escape and is consistent with JWST observations. More broadly, we conclude that the erosion of primary atmospheres typically does not preclude surface habitability, and frequently results in large surface water inventories due to the reduction of FeO by $H_2$.

Understanding the distribution of life in the Universe is inextricably linked to whether Earth's long-term habitability is a rare or common feature of habitable zone terrestrial planets. One potentially under-explored dimension of exceptionalness is the ways in which initial volatile inventories may shape subsequent geochemical evolution to preclude (or enhance) long-term habitability. The Kepler era of exoplanet exploration revealed two distinct types of planets of an intermediate size with no solar system analog: volatile-rich sub-Neptunes ($R > \sim 1.7 R_E$) and terrestrial-density super-Earths ($R \leq \sim 1.7 R_E$). While sub-Neptune bulk composition ($H_2$, $H_2O$, etc.) is a subject of ongoing investigation, and the possibility of different formation pathways for super-Earths and sub-Neptunes has not been discounted[1–5], there is evidence that these two planetary types derive from the same volatile-

rich parent population: super-Earths arise from the complete erosion of primary atmospheres, whereas sub-Neptunes have retained a substantial portion of their primary atmospheres[6]. The dominant mechanism for primary atmosphere erosion remains a topic of debate, but extreme ultraviolent (XUV) driven atmospheric escape and mass loss driven by interior heat flow both offer plausible explanations[7–12]. One important implication of this population-level understanding is that many—perhaps most—now-rocky exoplanets formed with substantial, long-lived primary atmospheres. This is broadly consistent with theoretical expectations that rapidly accreted terrestrial bodies ought to capture nebular gas[13,14].

Crucially, however, the abovementioned formation pathway for super-Earths is distinct from that of the Earth and other solar system

[1]Department of Earth and Space Sciences/Astrobiology Program, University of Washington, Seattle, WA 98195, USA. [2]NASA NExSS Virtual Planetary Laboratory, University of Washington, Seattle, WA 98195, USA. [3]NASA Ames Research Center, Moffett Field, CA 94035, USA. [4]Department of Earth and Planetary Sciences, ETH Zurich, Zürich, Switzerland. [5]NASA Hubble Fellowship Program Sagan Fellow, Earth and Planets Laboratory, Carnegie Institution for Science, Washington DC 20015, USA. [6]Department of Astronomy and Astrophysics, University of California, Santa Cruz, Santa Cruz, CA 95064, USA. ✉e-mail: joshkt@uw.edu

terrestrial planets, for which there is no evidence of prolonged initial H$_2$-rich envelopes, and limited evidence for nebula capture of volatiles[15,16]. Instead, the volatile endowments of solar system terrestrial bodies were sourced from chondritic material, perhaps with some small volatile contribution from planetesimals that formed beyond the snow line[17–19]. Recently, Young, et al.[20] argued that Earth's water inventory, oxygen fugacity, and core density can be explained by thermochemical equilibration between primordial H$_2$ and the rocky embryos that ultimately built Earth, but the extent to which this intriguing hypothesis can be reconciled with isotopic tracers of Earth's accretion has yet to be fully explored.

Distinct formation pathways for solar system terrestrials and exoplanets raise an important question: how might the presence of a long-lived primary atmosphere affect the subsequent habitability prospects of rocky exoplanets, given that solar system terrestrial bodies may have obtained their volatile endowments in an atypical manner? Now that atmospheric characterization of rocky planets around M-dwarfs is possible with JWST, an understanding of how primary atmospheres affect secondary atmosphere evolution is needed. There have been preliminary investigations on this topic[21,22]. In particular, Kite and Barnett[23] argued that XUV-driven loss of primary atmospheres will typically drag along high molecular weight volatiles, leaving behind a desiccated terrestrial planet, unless surface volatiles are replenished by mantle degassing. While instructive, these calculations are most applicable to highly irradiated (uninhabitable) planets and treat all non-hydrogen species as a single high molecular weight composite. Note that this escape process is inseparable from the thermal evolution of the post-accretional magma ocean; terrestrial planets are expected to form hot, and volatiles are partitioned between the partially molten mantle and atmosphere during early evolution. There has been no self-consistent investigation of how high molecular weight volatile element inventories partition and escape during primary atmosphere loss, or whether temperate terrestrial exoplanets that previously possessed a thick primary atmosphere would retain sufficient volatiles to sustain a biosphere.

Indeed, classical models of magma ocean thermal-climate-redox evolution are typically limited to H$_2$O and CO$_2$ (and occasionally O$_2$) bulk atmospheric compositions[24–28]. This assumption of a relatively oxidizing atmosphere is based on evidence for Earth's rapid differentiation and mantle oxidation after accretion[29,30], along with the absence of any evidence for a long-lived H$_2$-rich atmosphere[15]. Attempts to extrapolate these Earth-analog models to terrestrial exoplanets[26,28,31] typically ignore any H$_2$-rich initial atmosphere under the assumption that such atmospheres are too short-lived to affect subsequent evolution. In recent years, more generalized magma ocean models have been developed that can accommodate diverse redox chemistries and atmospheric compositions[32–35]. This body of work suggests a broader range of magma ocean atmospheres are possible, even for literal Earth twins. However, there has yet to be a self-consistent investigation of magma ocean evolution accommodating the reducing power of an initial H$_2$ envelope coupled to secular oxidation caused by XUV-driven escape, to investigate the transition from primary to secondary atmospheres.

Here, we present the first such self-consistent model of coupled magma ocean–redox–climate evolution. We are particularly interested in planetary volatile inventories post primary atmosphere loss, and the implied habitability prospects for terrestrial planets that formed with large nebular atmospheres. Planetary volatile inventories can significantly influence long-term habitability prospects of terrestrial planets. For example, planetary carbon inventories affect climate-stabilizing silicate weathering feedbacks[36,37] and even mantle dynamics and melt production[38]. Similarly, large water inventories affect redox evolution[28,39], resulting in abiotic oxygen-rich atmospheres that are not conducive to abiogenesis[40]. Large water inventories may also suppress carbon cycle feedbacks with implications for surface climate evolution[41]. If the loss of a primary atmosphere systematically modifies planetary volatile inventories, then the subsequent geochemical cycling of terrestrial exoplanets might be dramatically affected in observable ways, even billions of years after their primary atmospheres have been eroded. In what follows, we present a generalized, coupled atmosphere-interior evolution model to investigate the astrobiological legacy of primary atmospheres and to make testable predictions on the habitability prospects of rocky exoplanets for space-based observations.

## Results

### Modeling Approach

Figure 1 shows a schematic of the new model used in this study, PACMAN-P (PACMAN for Primary atmospheres). This is a significant upgrade to the original PACMAN model described in Krissansen-Totton and Fortney[42]. The subsections in the Methods section describe the individual components of the model, but broadly speaking we simultaneously solve for geochemical and thermal equilibrium as the magma ocean solidifies from the core-mantle boundary to the surface. At each time step, we find the multiphase equilibrium of all C, H, O, and Fe-bearing species between the atmosphere and the magma ocean. The resulting atmospheric species determine surface temperature, as calculated using a radiative-convective climate model that balances absorbed stellar radiation, outgoing longwave radiation, and internal heatflow. Internal heatflow is determined by parameterized mantle convection with temperature-dependent mantle viscosity, and driven by the heat of accretion, assumed radionuclides, and the latent heat of mantle solidification. Imposed stellar bolometric and XUV evolution– including a super luminous pre-main sequence–drives atmospheric escape, which is either XUV-limited or diffusion limited, depending on the composition of the upper atmosphere (computed via the climate model). Crucially, C-bearing and O-bearing species can be lost to space via hydrodynamic drag in the XUV-limited regime. We do not attempt to model atmosphere-interior interactions after the magma ocean has solidified, but we do continue to track stellar evolution and consequent atmospheric escape. To generate imminently testable predictions, we use TRAPPIST-1e and b[43] as case studies in all model calculations, but we note that qualitative model behavior is general to all temperate planets that form with substantial nebular atmospheres. Example calculations for other planetary systems (as well as Earth + Venus validation calculations) are presented in supplementary materials. A Monte Carlo approach is used to investigate sensitivity to initial volatile inventories, and uncertainties in stellar evolution, atmospheric escape, and other planetary parameters.

### TRAPPIST-1 simulations

Figure 2 shows the time evolution of TRAPPIST-1e for an approximately Bulk Silicate Earth (BSE)-like initial volatile endowment (i.e. no large H$_2$ envelope), an Earth-like mantle FeO abundance, and nominal point estimates for all other parameters (see Methods). Figure 2a shows the evolution of surface and mantle temperature, Fig. 2b shows the planetary energy budget consisting of OLR, ASR, and interior heatflow, Fig. 2c shows the solidification radius as it moves from the core-mantle boundary to the surface. Figure 2d shows the evolution of the oxygen fugacity of the magma ocean and growing solid mantle, Fig. 2e shows the evolution of atmospheric composition, and Fig. 2f–h, and i show inventories of Fe, H, C, and O respectively (and Figs. 3, 4, S1, and S2 follow the same layout). Mass is conserved except for H, C, and O-bearing species lost to thermal escape. The dashed vertical line in all subplots denotes the termination of the magma ocean i.e. when surface temperature drops below the solidus. After the termination of the magma ocean, only stellar evolution and atmospheric escape continue –there is no further chemical interaction between surface volatiles and the silicate interior, though solid mantle redox state does evolve with mantle temperature. The model is therefore conservative regarding habitability prospects since, in reality, surface volatile reservoirs will be

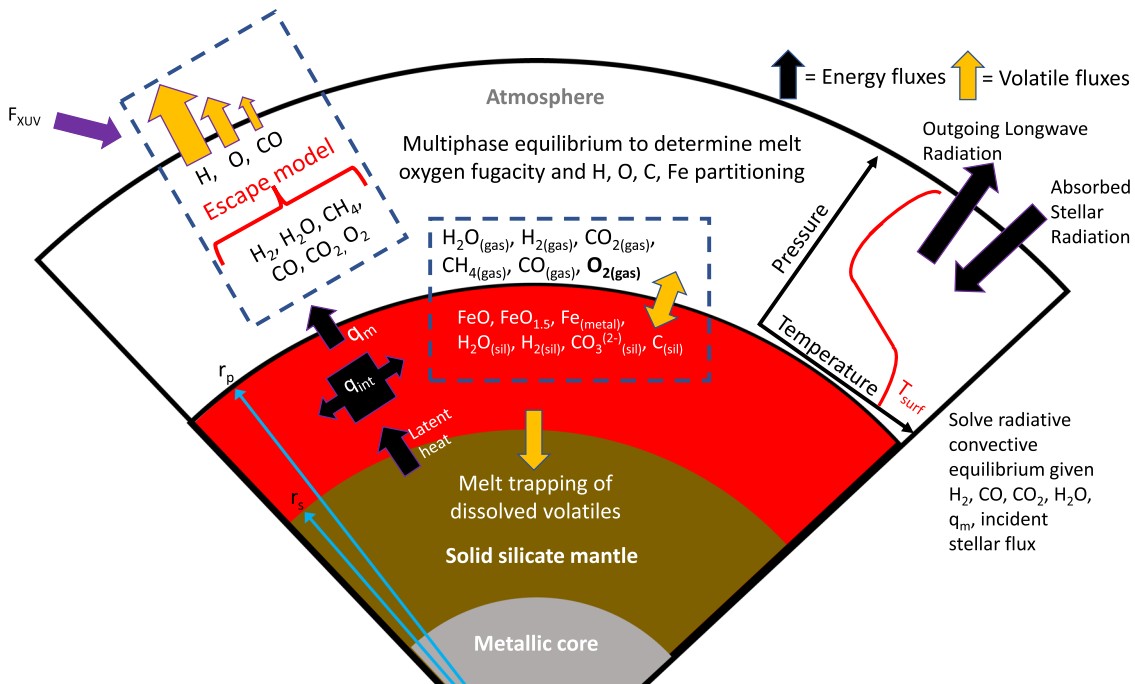

**Fig. 1 | Schematic overview of coupled redox-atmosphere-interior evolution model, PACMAN-P.** The model is initialized with a completely molten mantle, and the solidification radius, $r_s$, evolves from the core-mantle boundary to the surface, $r_p$, as the planet cools. At each timestep, we calculate multiphase thermochemical equilibrium between C, H, O, and Fe-bearing species to find the mantle redox state and the partitioning of volatiles between the magma ocean and the atmosphere. Surface climate, $T_{surf}$, is calculated using a radiative-convective model given atmospheric volatile inventories. Under highly reducing conditions created by large $H_2$ envelopes, the reaction of $H_2$ with the silicate magma ocean produces metallic iron, that is either sequestered in the core or remains in the mantle (two endmember assumptions). Atmospheric escape is either XUV-limited or diffusion limited depending on stratospheric composition, and under the former regime we calculate the hydrodynamic drag of O and CO.

replenished by magmatic degassing after magma ocean solidification. No metallic iron is produced in this comparatively oxidized scenario.

In this example calculation for TRAPPIST-1e, the atmosphere is dominated by CO and $CO_2$ throughout the planet's evolution due to the high solubility of water in silicate melts and comparatively low BSE-like H abundance (equivalent to a few Earth oceans). During the magma ocean stage, the atmosphere evolves from CO-dominated to $CO_2$-dominated, in part due to secular cooling, but also because H loss when the magma ocean is shallow (large solidification radius) oxidizes the melt-atmosphere system. The magma ocean ends after only a few ~$10^7$ years because there is insufficient water to maintain surface temperatures above the solidus; the planet remains in a runaway greenhouse state for longer, but there is insufficient greenhouse warming from $CO_2$ (with modest amounts of $H_2O$ and $H_2$) to keep the surface molten. The remaining water vapor is mostly lost to space, leaving behind a $CO_2$-$O_2$ atmosphere, where the $O_2$ is a consequence of H escape post magma ocean solidification (reduced atmospheric species like CO, $CH_4$, and $H_2$ are oxidized to $CO_2$ and $H_2O$). Atmospheric $H_2O$ is not lost completely because a cold trap is established as stellar bolometric luminosity falls. It is conceivable that water degassing post magma ocean solidification could transition this planet to a habitable state, as more than an Earth ocean is retained the in solid mantle and comparatively little C is lost to escape; whether surface liquid water can be stabilized depends on the balance of water degassing and atmospheric escape during the remaining pre-main sequence[44]. Suffice to say habitability is a possible outcome, but by no means guaranteed for this BSE-like initial condition.

Next, we consider the exact same scenario, except that TRAPPIST-1e is endowed with a substantial (0.1% planetary mass) initial $H_2$ envelope on top of the BSE-like volatile endowment (Fig. 3). Note that the additional C and O associated with a nebular atmosphere (assuming solar C:H and O:H) would not dramatically alter our assumed initial

carbon and free oxygen inventories[45]. In this case, we make the endmember assumption that all metallic iron formed by the reduction of FeO is retained in the mantle, but results are qualitatively similar for the opposite case where all metallic iron is sequestered in the core (see below). In Fig. 3, the large $H_2$ endowment results in a much longer-lived magma ocean (a few ~$10^8$ years) due to the strong greenhouse warming from the $H_2 + H_2O$ atmosphere. The atmosphere is $H_2$-dominated during the magma ocean phase (Fig. 3e), but later transitions to water-dominated due to oxidation from H loss and water exsolution from the magma ocean. Exsolved water is predominantly a byproduct of the reduction of the mantle by the nebular atmosphere, FeO + $H_2$ -> $H_2O$ + Fe. Note that atmospheric H increases during rapid stages of magma ocean solidification even though total planetary H is decreasing because the exsolution of water melt is more rapid than H escape, but during the final (slow) stages of solidification, both planetary H and atmospheric H decline. A reducing, $H_2$-dominated ($CH_4$-rich) atmosphere persists briefly after magma ocean solidification, and a solidified TRAPPIST-1e transitions to a habitable state (Fig. 3a) with a deep (~20 km) surface ocean. Shortly after, cumulative H escape flips the redox balance of the surface to net oxidized ($H_2O$-$CO_2$-$O_2$) and a $CO_2$-$O_2$ dominated atmosphere persists as most water is condensed on the surface (Fig. 3e). While some carbon is lost to escape during the pre-main sequence (Fig. 3h), a large $CO_2$ inventory remains that could be sequestered as carbonates and enable a silicate weathering thermostat.

We contrast this habitable outcome for TRAPPIST-1e to the magma ocean evolution of TRAPPIST-1b (Fig. 4) with the identical initial composition (0.1% nebular atmosphere, Earth-like initial FeO, etc.), and the same endmember assumption whereby all metallic iron remains in the mantle. Here, the magma ocean persists until virtually all fluid H is lost to space (Fig. 4e) due to higher received bolometric and XUV fluxes. Essentially all planetary carbon is lost to space

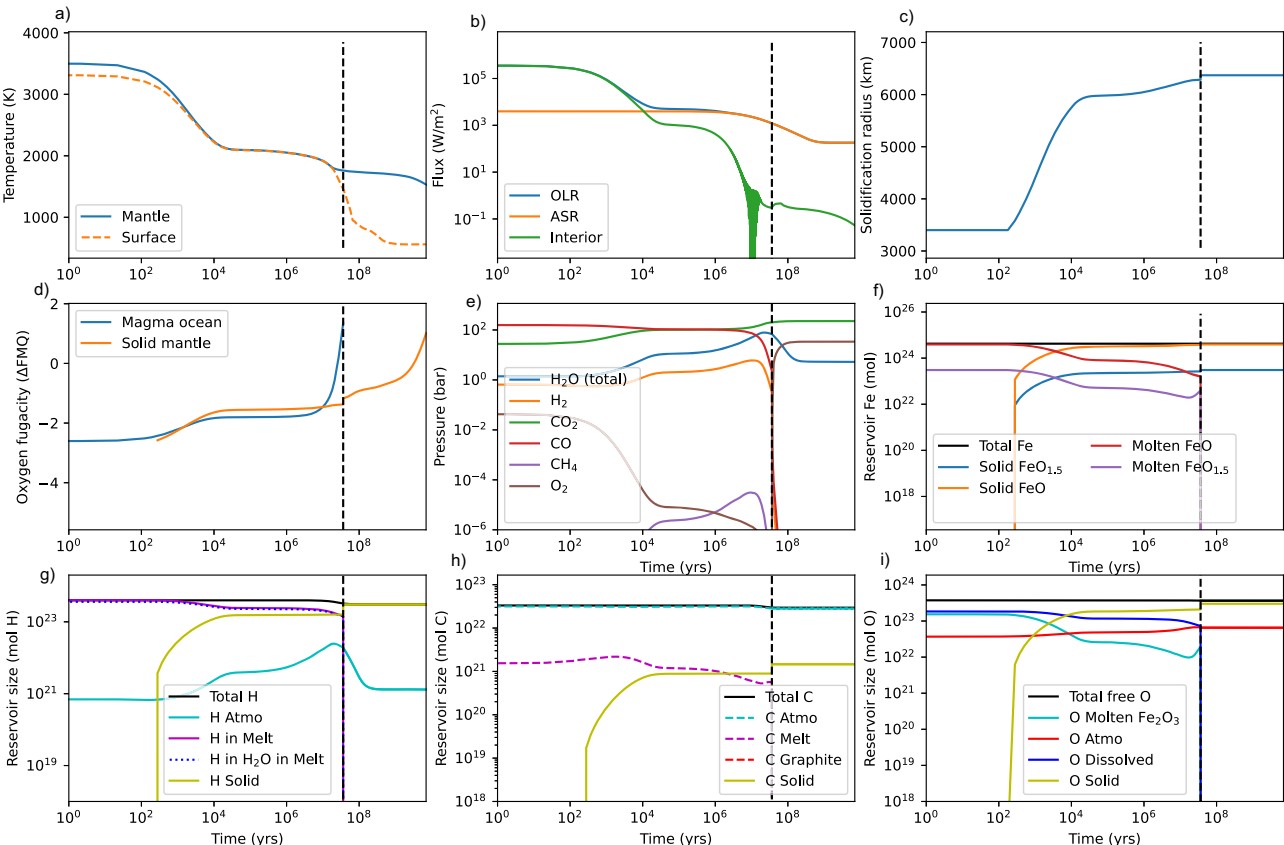

**Fig. 2 | Time-evolution of TRAPPIST-1e, with Earth-like initial volatile inventories (no substantial primary atmosphere) from post-accretion magma ocean to present (8 Gyr).** Subplot (**a**) denotes the time-evolution of surface (orange) and mantle potential temperature (blue), (**b**) denotes the evolution of outgoing long-wave radiation (OLR, blue), absorbed shortwave radiation (ASR, orange), and interior heatflow (green), and subplot (**c**) shows the evolution of the magma ocean solidification front from the core-mantle boundary to the surface. Subplot (**d**) shows solid mantle (orange) and magma ocean redox (blue) relative to the Fayalite-Quartz-Magnetite (FMQ) buffer, (**e**) shows the evolution of atmospheric composition including $H_2O$, $H_2$, $CO_2$, CO, $CH_4$, and $O_2$. Subplot (**f**) denotes iron speciation in both the magma ocean and the solid silicate mantle−in this oxidizing Earth-analog case no metallic iron is produced. Subplot (**g**) shows both solid and fluid reservoirs of H; total dissolved hydrogen (purple) and hydrogen dissolved as $H_2O$ (blue-dotted) are essentially identical in this oxidized scenario where dissolved $H_2$ is minimal. Subplot (**h**) denotes solid and fluid reservoirs of C and subplot (**i**) denotes solid and fluid reservoirs of free oxygen, including oxygen bound to ferric iron, atmospheric species, and O in volatiles dissolved in the melt ($H_2O$, $CO_2$) reservoirs, respectively. Vertical dashed black lines show the termination of the magma ocean, which takes $\sim 4 \times 10^7$ years in this case. The short duration of the magma ocean is attributable to the small H inventory, which in turn means limited time for the hydrodynamic drag of C and O; substantial C and O inventories are retained post-magma ocean solidification, and subsequent habitability is not precluded. Note that we do not explicitly model mantle-atmosphere exchange after magma ocean solidification, only escape and stellar evolution.

(Fig. 4h), consistent with previous predictions for highly irradiated terrestrial planets[23]. After the magma ocean ends, TRAPPIST-1b is left with a comparatively tenuous (~few bar) $O_2$ atmosphere that would likely be highly susceptible to non-thermal escape[46–49], which we do not consider here. Carbon is lost more completely than oxygen due to the low solubilities of C-bearing species in silicate melts. The small mantle volatile content implies limited potential for replenishing the atmosphere of TRAPPIST-1b by volcanic degassing. Results are qualitatively similar if all metallic iron is assumed to be sequestered in the core (Fig. S7), except that the final mantle redox state is around FMQ + 4 as opposed to FMQ-3 and a more substantial $O_2$ atmosphere remains.

**Monte Carlo Analysis**

To systematically map out the parameter space of possible evolutionary pathways from primary to secondary atmospheres, we conducted Monte Carlo analyses of the evolution of TRAPPIST-1e (and b) over a broad range of initial conditions, escape, and stellar evolution parameters (see Methods). We focus on initial nebular atmospheres (yielding endogenous water) as opposed to water-rich bulk compositions due to ice accretion to be conservative with respect to atmospheric retention prospects. Figure 5 shows the outcome of these

Monte Carlo calculations as a function of initial $H_2$, which ranges from an order of magnitude below Bulk Silicate Earth (BSE) all the way up to a substantial (1–2 wt% H) nebular atmosphere. The first two columns denote TRAPPIST-1e and the last two columns denote TRAPPIST-1b outputs; the two columns for each planet represent endmember cases whereby all metallic iron is sequestered in the core, and all metallic iron remains in the silicate mantle, respectively. The four rows show final total atmospheric pressure, partial pressures of surface volatiles, final carbon inventories, and final hydrogen inventories, respectively.

The outputs for TRAPPIST-1e reveal that, after 8 Gyr of evolution, a large surface inventory of water remains for planets with substantial (>BSE) initial H inventories. A Venus-like $CO_2$-dominated atmosphere remains for smaller initial H inventories. The median final planetary carbon inventory for TRAPPIST-1e is comparable to BSE abundances (i.e. most carbon is typically retained and not lost via hydrodynamic drag). We do not consider initial H inventories beyond ~$10^{23}$ kg (~2 wt%) since TRAPPIST-1e often retains H-rich atmospheres for its entire 8 Gyr evolution in these cases, in conflict with observations[50]. For TRAPPIST-1b, a thinner, $O_2$-dominated atmosphere is the most common outcome across a broad range of initial H inventories. Final planetary carbon inventories are typically orders of magnitude lower than for TRAPPIST-1e due to more protracted hydrodynamic drag and

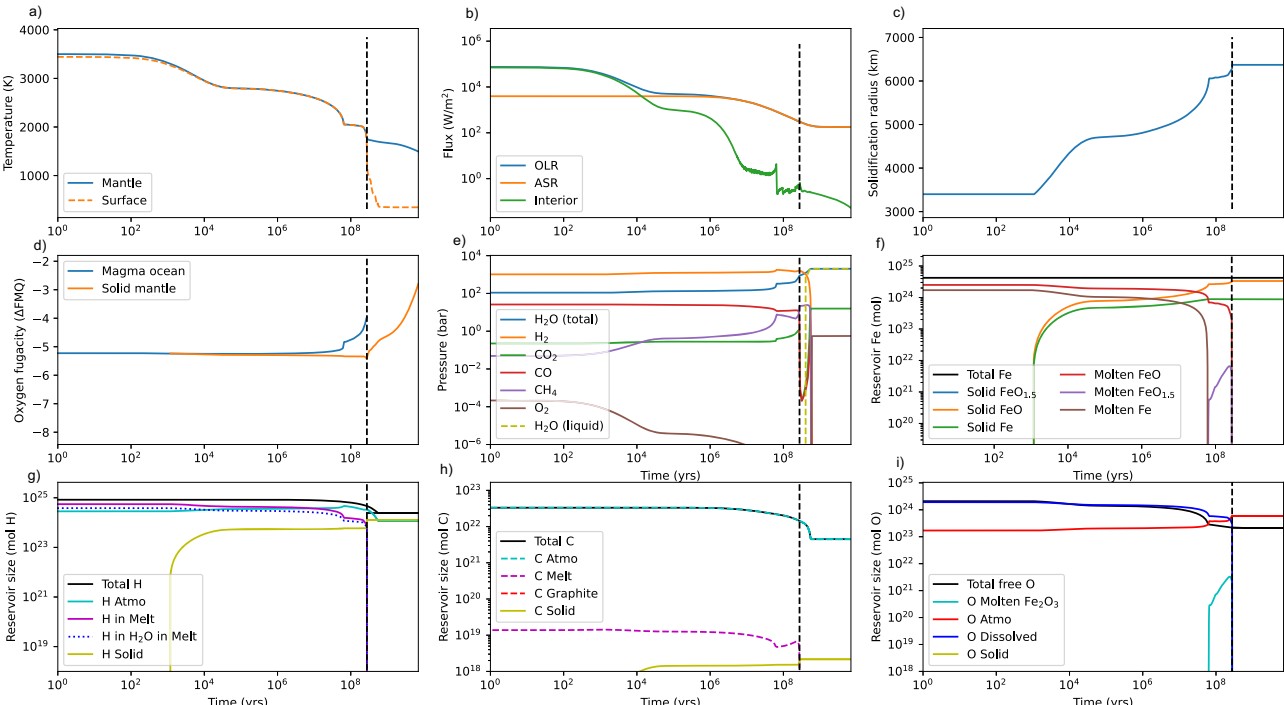

**Fig. 3 | Time-evolution of TRAPPIST-1e, with a large initial $H_2$ envelope from post-accretion magma ocean to present (8 Gyr).** Subplot (**a**)–(**i**) are identical to those in Fig. 2. In this case, an $H_2$-dominated atmosphere persists for the duration of the magma ocean (**e**). Vertical dashed black lines show the termination of the magma ocean, which takes $\sim 3 \times 10^8$ years in this case. The longer duration of the magma ocean is attributable to the large H inventory, which in turn means more time for the hydrodynamic drag of C and O; nonetheless substantial H (**g**), C (**h**), and O (**i**) inventories are retained post magma ocean solidification, and subsequent

habitability is not precluded. In fact, reaction of $H_2$ with the silicate magma ocean generates abundant $H_2O$ that ultimately forms a deep surface ocean (~7 Earth oceans). In this particular case, we make the endmember assumption that all metallic iron generated by the reaction of FeO and $H_2$ remains in the mantle, and so metallic species are present in (**f**). The atmospheric outcomes are qualitatively similar if all metallic iron is assumed to be sequestered in the core (see Fig. S6). Note that we do not explicitly model mantle-atmosphere exchange after magma ocean solidification, only escape and stellar evolution.

higher XUV fluxes during the pre-main sequence. TRAPPIST-1b is often completely desiccated after 8 Gyr of evolution with either no surface water (initial H < $3 \times 10^{21}$ kg) or limited water vapor in a dense $O_2$ atmosphere, and water in the silicate mantle that is an order of magnitude less than BSE.

Broadly speaking, the Monte Carlo results shown in Fig. 5 confirm what was suggested by the illustrative cases above (Figs. 2–4). The reaction of a hydrogen-rich primary atmosphere with the silicate mantle produces water rich composition for both planets (see also Kimura and Ikoma[51], Kimura and Ikoma[52]). The comparatively short runaway greenhouse phase of TRAPPIST-1e (few 100 Myr) means that most of this water is retained, alongside sizeable carbon inventories. A silicate weathering thermostat and temperate, habitable conditions are not precluded on TRAPPIST-1e for initial H inventories equal to or exceeding BSE.

In contrast, because TRAPPIST-1b resides interior to the runaway greenhouse limit throughout its evolution, virtually all the surface water produced by the reduction of FeO with $H_2$ is lost to space, sometimes leaving behind a thin $O_2$-dominated atmosphere. The top subplots of Fig. 5 which show final, total surface volatile pressure (bar) are overplotted with a plausible estimate of the maximum atmospheric loss to non-thermal processes[46,47], which are not explicitly considered here. For TRAPPIST-1b, final surface pressure is comparable to or below this threshold for a wide range of initial H inventories, thereby providing a natural explanation for the apparent airlessness (or thin atmospheres) of TRAPPIST-1b and c suggested by thermal emission observations with JWST[53,54]. This theoretical prediction of surface H, C, and O depletion is qualitatively consistent with Kite and Barnett[23]. In contrast, a substantial atmosphere is predicted for TRAPPIST-1e regardless of initial H inventory.

The non-monotonic behavior of final pressure vs. initial hydrogen can be understood intuitively. At least some hydrogen is needed for the XUV-driven drag of heavier species, otherwise C and O-bearing species cannot escape thermally. Increasing initial H on highly irradiated planets first decreases final C and O as these species are dragged along in the hydrodynamic wind[23]. However, adding more $H_2$ also liberates more O via FeO reduction as described above, and so at very high initial $H_2$ inventories not all O is typically lost. Moreover, the high mixing ratios of $O_2$ and $H_2O$ mean that most of the absorbed XUV radiation drives H loss and the drag of O, whereas C is somewhat shielded from complete thermal escape by its low mixing ratio.

## Discussion

The calculations described above make clear predictions for the atmospheric retention and habitability prospects for terrestrial planets around low mass stars. TRAPPIST-1e is used as an illustrative example since, in many respects, it is a "worst case scenario" for habitability: its M8 host star underwent a protracted pre-main sequence subjecting its planetary system to high bolometric and XUV fluxes for hundreds of millions of years (planets around early M-dwarfs experience less cumulative XUV). Moreover, TRAPPIST-1e is closest to the inner edge of the habitable zone[55] and thus underwent a longer runaway greenhouse during the pre-main sequence compared to f and g. Consequently, the calculations have implications for habitability prospects for Earth-sized planets around M-dwarfs more broadly. Indeed, we include example calculations applying our model to other M-dwarf hosted planetary systems (LP 890-9 and Proxima Centauri) in the supplementary materials and find very similar results. We conclude that habitability is not precluded by formation with, and subsequent loss of, an H-rich primary atmosphere, or by the extreme XUV

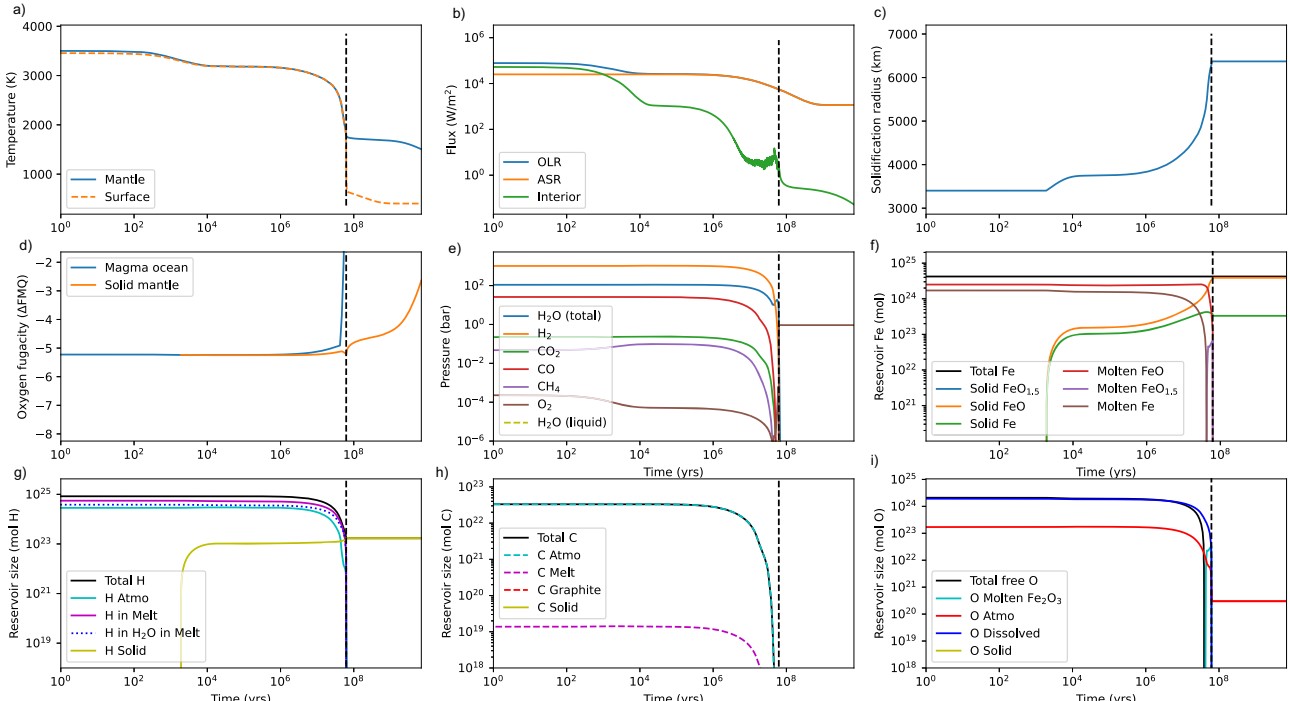

**Fig. 4 | Time-evolution of TRAPPIST-1b, with a large initial $H_2$ envelope (i.e. the same initial volatile inventories as for TRAPPIST-1e in Fig. 3) from post-accretion magma ocean to present (8 Gyr).** Subplot (**a**)−(**i**) are identical to those above. In this case, an $H_2$-dominated atmosphere transitions to an $H_2O$-dominated atmosphere as H is lost to space and the magma ocean degasses. Unlike for TRAPPIST-1e, virtually all atmospheric H (**g**) and C (**h**) are lost to space due to higher experienced XUV fluxes. A small (few bar) $O_2$ atmosphere is retained that would be vulnerable to non-thermal escape mechanisms. As in Fig. 3, we make the end-member assumption that all metallic iron generated by the reaction of FeO and $H_2$ remains in the mantle, and so metallic iron is present in (**f**). The atmospheric outcomes are qualitatively similar if all metallic iron is assumed to be sequestered in the core (see Fig. S7). Note that we do not explicitly model mantle-atmosphere exchange after magma ocean solidification, only escape and stellar evolution.

environment created by pre-main sequence M-dwarfs. Our results are qualitatively consistent with planetary population synthesis models that similarly predict water-rich temperate, terrestrial planets around M-dwarfs from the reaction of nebular $H_2$ with silicates[51]. More broadly, these findings suggest temperate planets around M-dwarfs remain compelling targets for atmospheric characterization and astro-biological investigation.

Indeed, these simulations permit a stronger conclusion than that of Krissansen-Totton[56], which found that the airlessness of TRAPPIST-1b and c does not imply e and f are airless. Here, the calculations in Fig. 5 support the affirmative statement that TRAPPIST-1e is likely to have retained an atmosphere despite vigorous hydrodynamic escape of volatiles in the high XUV environment of the TRAPPIST-1 star. By extension, f and g are even less likely to be airless for the same reason (see Supplementary Fig. S18), although for cooler planets nightside condensation of volatiles must also be considered[57,58]. This conclusion holds for a broad range of initial H inventories all the way from Earth-like, up to maximum initial H-envelope consistent with lack of $H_2$-rich atmosphere today (Fig. 5).

With that said, the simulations described above have several caveats that we will now consider. First, if cumulative non-thermal escape fluxes are much larger than most current estimates, then TRAPPIST-1e—and other temperate planets around late M-dwarfs—may be airless regardless of initial volatile endowment. In Fig. 5 the assumed shaded upper limit(s) to non-thermal escape is based loosely on esti-mates in Dong, et al.[46], who find ion escape rate of 0.3−2 bar/Gyr for TRAPPIST-1b and 0.1−0.2 bar/Gyr for TRAPPIST-1e assuming $CO_2$-dominated atmosphere, as well as Garcia-Sage, et al.[47], who report an upper limit to ion escape rates from Proxima Centauri b of approxi-mately 6 bar/Gyr assuming the highest possible thermosphere tem-perature and completely open magnetic field lines. However, for H-O

atmospheres Dong, et al.[48] estimated ion escape rates around TRAPPIST-1g of up to 20 bar/Gyr, ostensibly due to the lack of radiative cooling from $CO_2$. Substantial impact erosion rates have also been suggested for TRAPPIST-1[59], although these are seemingly improbable because the fragile resonance of the TRAPPIST-1 system puts upper limits on the mass flux since the formation of the resonant chain[60]. In any case, given the results described above, if the outer planets of the TRAPPIST-1 system are shown to be airless, then this would suggest either non-thermal loss rates higher than expected in C-bearing atmospheres (e.g. Ref. 48) or initial volatile inventories substantially smaller than that of the Earth[42,61]. While thermal escape rates far larger than our parametrization permits have been suggested[62], these results must be reconciled with atmospheric retention in the early solar sys-tem, and also disagree with escape models that incorporate detailed atomic line cooling, which instead predict secondary atmospheres that are robust to thermal escape under high XUV fluxes[63].

We similarly emphasize that post magma ocean evolution is not fully modeled in this study; after magma ocean solidification we con-sider only continued stellar evolution and thermal escape, to enable fair comparison of final outcomes of model runs with different dura-tion magma oceans. In reality, volatile cycling between the mantle and interior via degassing and weathering may occur[42,56]. Importantly, scenarios where substantial $H_2O$ and C inventories are retained after 8 Gyr evolution do not necessarily guarantee habitability due to the omission of atmosphere-interior exchange processes. One potential impediment to habitability could be the formation of dense, high-pressure ices at the base of ocean that impede carbon cycling and nutrient exchange with the rocky interior[64]. For the largest initial $H_2$ inventories we consider, the partial pressure of water at the surface of TRAPPIST-1e at the end of our calculations is 2−9 kbar (Fig. 5). Given the final surface temperature distribution for TRAPPIST-1e (Fig. S8)

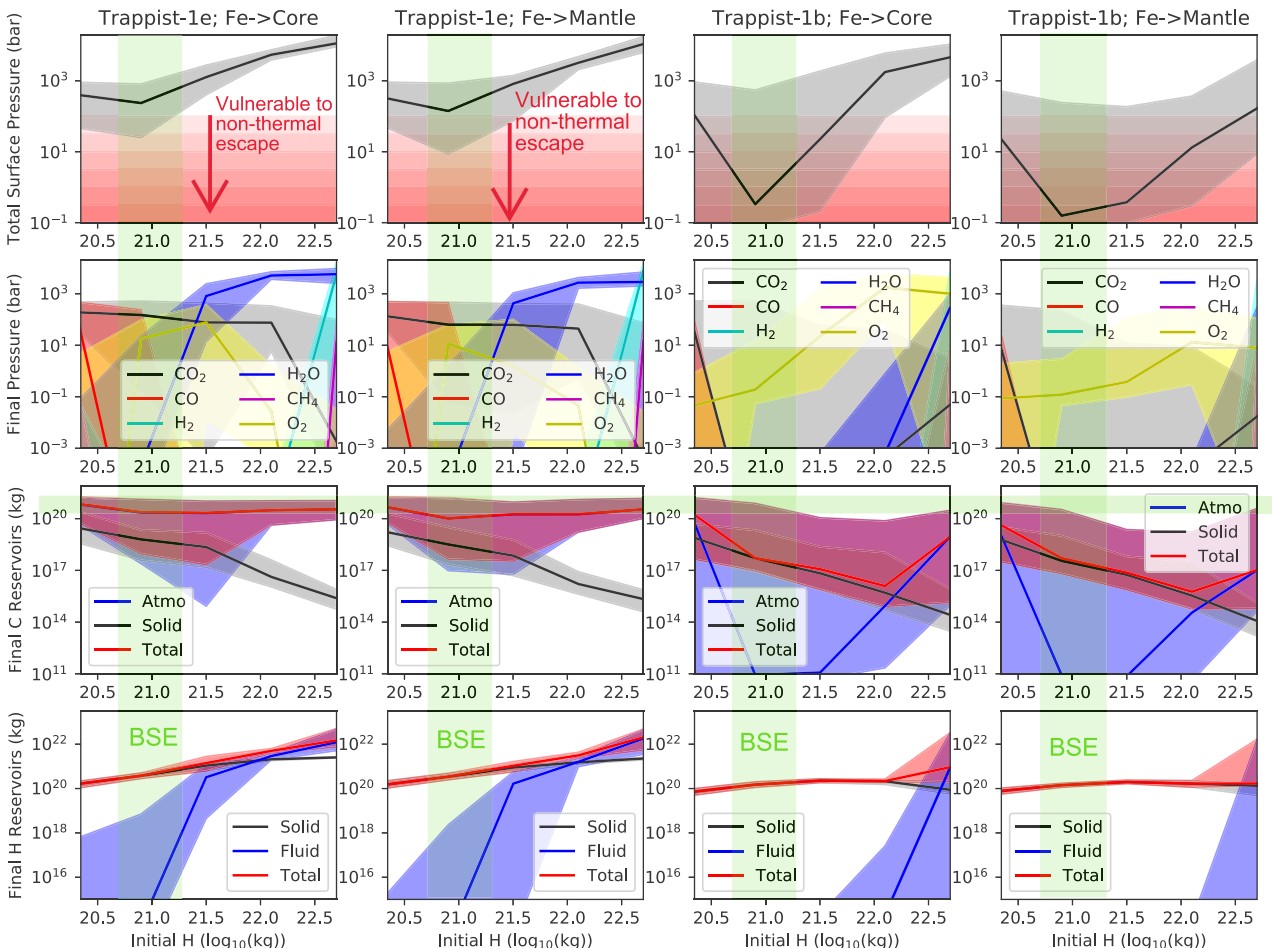

**Fig. 5 | Monte Carlo evolutionary calculation showing outcomes for TRAPPIST 1e (left two columns) and TRAPPIST-1b (right two columns) after 8 Gyr of coupled atmosphere-interior-redox evolution, as a function of initial H endowment.** Rows denote total surface pressure (bar), partial pressures of surface volatiles (bar), final atmosphere and interior carbon inventories (kg C), and final atmosphere and interior hydrogen inventories (kg H), respectively. The two columns for each planet denote endmember cases whereby all metallic iron is sequestered in the core (left), and all metallic iron remains in the silicate mantle (right). Solid lines and shaded regions denote median model outputs and 1-sigma confidence intervals, respectively. The green-shaded ranges denote approximate modern Bulk Silicate Earth (BSE) volatile abundances, and the red gradients show total surface volatile inventories increasingly vulnerable to non-thermal escape— final volatile inventories in the red-shaded region represent model runs that could result in airless planets. Broadly speaking, TRAPPIST-1e is expected to retain substantial volatiles and habitability is not precluded, whereas TRAPPIST-1b is likely to be left with a comparatively thin $O_2$ atmosphere susceptible to complete erosion via non-thermal loss.

high-pressure ices are not predicted to form at the base of this ocean[65]. Of course, if TRAPPIST-1e accreted large amounts of water ice[66] then larger surface water inventories are possible. But the accretion of nebular gas and subsequent loss of the primary atmosphere does not, on its own, yield too much water to preclude habitability. More broadly, our results indicate that sufficient surface and interior volatile reservoirs are retained to permit subsequent biogeochemical cycling. A more detailed exploration of atmosphere-interior evolution post magma ocean is a topic for future work, although we note that the possibility of degassing of mantle H, C, and O would only help to replenish surface inventories and buffer against atmospheric escape.

Secondly, our atmosphere-interior-redox evolution model is limited to C, H, O, and Fe-bearing species interacting with a silicate mantle. This simplification omits the influence of other volatiles and redox-sensitive species. For example, the redox sensitivity of Si is neglected; under highly reducing conditions, $H_2$ may reduce silica to produce additional water and metallic Si via the reaction: $SiO_2 + 2H_2 \rightarrow 2H_2O + Si_{(metal)}$. However, the inclusion of metallic Si is unlikely to significantly modify outcomes for the range of initial H inventories considered in this study. Schlichting and Young[67] performed a core-silicate-atmosphere equilibrium calculation for sub-Neptunes and find

that mantle $SiO_2$ is relatively insensitive to H inventory, and that metallic Si constitutes less than 0.01% of total metallic phases across all H inventories considered. We similarly neglect Si-bearing volatiles and their effect on thermal and redox evolution. While silicate vapor is unlikely to be important for the temperatures considered here[68], silane ($SiH_4$) is a potentially substantial atmospheric constituent for magma ocean worlds with large H inventories[69]. Specifically, the reaction of $H_2$ and $SiO_2$ can generate $SiH_4 + H_2O$ at surface temperatures in excess of ~1000 K[69]. While a full exploration of the effect of Si-bearing volatiles on planetary redox evolution is a subject for future work, we can investigate sensitivity to $H_2 \rightarrow H_2O$ conversion by repeating calculations for a broad range of initial FeO contents. Supplementary Fig. S10 and S11 repeats our nominal calculations for initial FeO contents varying from 0.02 to 0.2 mantle mole fraction and demonstrates that the conversion of $H_2$ to water is efficient in all cases; this suggests that the formation of water and silane is unlikely to dramatically change the results presented here; silane will be similarly photo-dissociated and H lost to space as the Si-H bond strength is comparable to the C-H bond strength in $CH_4$[70,71].

Note also that N- and S-bearing species are omitted from the simulations, except for an assumed fixed $N_2$ background in climate

calculations. Adding additional volatile species would not directly influence retention/escape of C, H, and O-bearing species, but future work ought to explore how N and S inventories are shaped by an extended H-rich atmosphere and pre-main sequence phase, and how the resultant silicate and atmospheric reservoirs might differ from BSE abundances as well as the implications for habitability e.g.[72,73]. Qualitatively, it is reasonable to expect similar outcomes for N as C given their insolubility in silicate melts, comparable cosmochemical abundances, and molecular weights.

The simulations described above do not permit C, H, and O-bearing species to partition into metallic iron. This does not substantially affect the endmember case whereby all metallic iron remains in the silicate mantle because all volatiles that partition into metallic phases remain in the mantle for subsequent degassing. However, if C-H-O partition into metallic phases and the metallic iron is permanently sequestered in the core, then this could diminish prospects for atmospheric retention and habitability. We instead argue that after accretionary core formation is complete (a process which we do not model here) volatile partitioning into metallic phases is minimal for the parameter space we consider. Dissolved carbon mole fractions in reducing silicate melts are typically low, around $10^{-7}$ in most of our model runs, but here we will suppose they are ~$10^{-5}$ to account for the contribution from neglected dissolved reduced species[74]. In our simulations, the metallic iron to ferrous iron mole fractions in the molten mantle are usually low, except for the highest initial H inventories, where they exceed 1. Thus, supposing a metal partition coefficient for carbon of $D_C = 100$[75,76], then the amount of carbon that partitions into metallic phases in equilibrium is $10^{-5} \times 100 \times$ (moles metallic Fe). For an Earth-like mantle FeO endowment (0.06 mole fraction), and half of that FeO reduced to metallic Fe by the nebular atmosphere, we have $10^{-5} \times 100 \times (0.03 \times 6.7 \times 10^{25}$ mol mantle) = ~$2 \times 10^{21}$ mol C sequestered in metallic phases after core formation. This is a small fraction of the atmospheric inventory, which is around $10^{22}$–$10^{23}$ mol C (Fig. 5). The high abundances of carbon-bearing species detected in warm sub-Neptune atmospheres such as TOI 270d[77,78] also argues against C being mostly lost to metallic phases, even under strongly reducing conditions. For hydrogen, Schlichting and Young[67] report only 1-10% of total planetary hydrogen partitions into the metallic core for temperatures ranging from 3000 to 5000 K, a deficit that's unlikely to affect the broad conclusions of this paper. Other work has similarly concluded a low core H content for Earth[79]. Of course, a more complete exploration of this topic would require an explicit exploration of volatile portioning during N-body growth and core formation (e.g. Refs. [80], [81]) with co-evolving climate and atmospheric escape; this ambitious undertaking is left for future work.

To conclude, for habitable zone planets, the transition from sub-Neptune to terrestrial planet via XUV-driven escape of a primary atmosphere typically does not strip the planet of high molecular weight volatiles, and instead is likely to leave behind large surface water inventories. This improves the habitability prospects for terrestrial plants around late M-dwarfs that accrete primary atmospheres. Consistent with previous calculations, the transition from sub-Neptune to terrestrial planets for highly irradiated, Venus-analogs is likely to strip high mean molecular weight volatiles and leave behind thin atmospheres (and small mantle volatile inventories) that are susceptible to erosion via non-thermal escape processes. The two conclusions above provide a natural explanation for the apparent airlessness or thin $O_2$-$CO_2$ atmospheres of TRAPPIST-1b and c favored by JWST observations. In contrast, the same model predicts that TRAPPIST-1e, f, and g are unlikely to be airless rocks given plausible initial inventories and thermal escape histories. If all the planets in the TRAPPIST-1 system are airless despite the predictions above, then this would strongly suggest either unexpectedly large non-thermal escape fluxes or initial volatile endowments much smaller than that of the Earth.

## Methods

### Radiative-convective climate model

The open-source radiative transfer code petitRADTRANS[82] was used to compute surface temperature as a function of atmospheric composition. Specifically, we precomputed outgoing longwave radiation (OLR) fluxes as a function of surface temperature, stratospheric temperature, $CO_2$, CO, $H_2$, and $H_2O$ inventories. In addition to line lists for $H_2O$, $CO_2$, and CO[83], we include continuum opacities for $H_2O$–$H_2O$, $CO_2$–$CO_2$, $H_2$–$H_2$, and CO-CO collisionally induced absorption[82]. Sensitivity tests show that $H_2$–He and $H_2O$–$CO_2$ CIA can be neglected for the atmospheres we consider, i.e. their additions to opacity are small compared to other sources of error. Given the lack of CO-CO CIA data, we use $N_2$–$N_2$ CIA as an analog based on isoelectronicity and identical molecular weights. Following previous magma ocean studies[28,32,84], a dry adiabat, to moist adiabat, to isothermal temperature structure was assumed; the temperature profile transitions from dry adiabat to moist adiabat when the partial pressure of water vapor drops below saturation, and the atmosphere transitions to isothermal when the temperature reaches the planetary skin temperature. Note that water is the only condensable species for the temperature ranges relevant to this study. For surface temperatures below the critical point of water (647 K) water partitions into atmospheric and condensed surface reservoirs as specified by saturated vapor pressure. At each timestep in the evolutionary model, we solve for the surface temperature that balance OLR—which itself depends on surface volatile inventories and atmospheric temperature profile—, heating from the interior, $q_m$ (described below) and absorbed shortwave radiation (ASR):

$$OLR(T_{surf}, \varsigma_{H_2O}, \varsigma_{H_2}, \varsigma_{CO_2}, \varsigma_{CO}, T_e) = q_m(T_{surf}, T_p, r_s) + ASR(t) \quad (1)$$

Here, ASR evolves with bolometric luminosity (see below), modulated by planet-star separation and assumed albedo. Albedo is a fixed parameter that is sampled broadly in our Monte Carlo analysis. Attempting to model the complex cloud microphysics, aerosol properties, atmospheric circulation patterns, and climate system feedbacks that control planetary albedo across the primary-to-secondary atmospheric transition would be a challenging undertaking that would require a hierarchy of computational models including 3D GCMs. Instead, we take a conservative approach with respect to the duration of magma ocean solidification and assume a low albedo range (0-0.2). This range is consistent with what is expected for cloud-free runaway greenhouse atmospheres on planets around late M-dwarfs[85,86], but also maximizes the duration of the runaway greenhouse compared to cloudy scenarios, and by extension the longevity of hydrodynamic escape. We later consider sensitivity tests with higher albedo values and find conclusions unchanged (see Supplementary Materials). The radiative effects of methane are neglected since methane abundances are only significant in $H_2$-dominated atmospheres, for which the greenhouse warming of $H_2$ typically dominates. We assume a fixed 1 bar background of $N_2$, which is expected to have a minimal impact on climate evolution for typical (10–1000 s of bar) surface inventories of C-, H-, and O-bearing species. Note that there are sometimes rapid variations in interior heatflow (e.g. Figure 2 around ~$10^7$ years) that are numerical artefacts attributable to an imperfect climate grid. By assuming an adiabat in the deep atmosphere, we neglect the possibility of a transition to a deep radiative zone[87]. This possibility is instead explored in supplementary materials and is found unlikely to affect qualitative conclusions. Additionally, more rapid magma ocean solidification would trap more volatiles in the mantle where they are shielded from early XUV fluxes and are available for later degassing – our climate model is therefore conservative on atmospheric loss. Finally, we conducted sensitivity tests with an independent, alternative climate model, *Clima*[40], and found qualitatively similar conclusions.

## Magma ocean thermal evolution

All models are initialized with a completely molten mantle, and the subsequent thermal evolution of the magma ocean is governed by energy balance:

$$\rho_m c_p V_m \frac{dT_p}{dt} = \rho_m V_m Q_r + Q_c + \rho_m 4\pi r_s^2 \Delta H_f \frac{dr_s}{dt} - q_m 4\pi r_p^2 \quad (2)$$

Here, $V_m = 4\pi(r_p^3 - r_s^3)/3$ is the volume of the molten mantle, $\rho_m$ is the average density of the mantle, $Q_r$ is radionuclide heat production per unit mass, $r_p$ is planetary radius, $\Delta H_f$ is the latent heat of fusion of silicates, $r_s$ is the solidification radius, $c_p$ is the specific heat of silicates, $Q_c$ is the heatflow from the metallic core[28], $T_p$ is mantle potential temperature, and $dr_s/dt$ is the time-evolution of the solidification radius. The heatflow from the interior to the atmosphere, $q_m$, is calculated using a 1-D convective parameterization, with temperature-dependent magma ocean viscosity, $\nu(T_p)$:

$$q_m = \frac{k(T_p - T_{surf})}{(r_p - r_s)} \left(\frac{Ra}{Ra_{cr}}\right)^\beta \quad (3)$$

Where $T_{surf}$ is surface temperature, and $k$ is thermal conductivity. The Rayleigh number, $Ra$, is given by

$$Ra = \frac{\alpha g(T_p - T_{surf})(r_p - r_s)^3}{\kappa \nu(T_p)} \quad (4)$$

Here, $\kappa$ is thermal diffusivity, $Ra_{cr}$ is the critical Rayleigh number, $g$ is surface gravity, and $\alpha$ is thermal expansivity. These equations continue to govern the thermal evolution of the mantle after the magma ocean has solidified, except with $dr_s/dt = 0$ (see below). The temperature-dependent mantle viscosity parameterization is identical to that of Krissansen-Totton, et al.[28] and smoothly transitions from low viscosity at high melt fractions to a solid-like "mush" at low melt fractions.

## Magma ocean geochemical evolution

At every timestep in the evolutionary model, we simultaneously solve a coupled system of equations describing the partitioning of C, H, O-bearing volatiles between atmosphere, molten silicate mantle (and metallic) phases, melt oxygen fugacity, iron speciation, and surface temperature. This system of equations is iteratively solved simultaneously with the radiative-convective climate model to self-consistently determine redox, climate, and volatile speciation at each time-step. Gas speciation is determined by the following equations:

$$\ln\left(\frac{K_1}{fO_2^{0.5}}\right) + \ln(fH_2O) = \ln(fH_2) \quad (5)$$

$$\ln\left(\frac{K_2}{fO_2^{0.5}}\right) + \ln(fCO_2) = \ln(fCO) \quad (6)$$

$$\ln\left(\frac{K_3}{fO_2^2}\right) + \ln(fCO_2) + 2\ln(fH_2O) = \ln(fCH_4) \quad (7)$$

Here, $K_1$, $K_2$, and $K_3$ are temperature-dependent equilibrium constants[88], and $fX$ represents the fugacity of volatile species $X$ (bar).

We adopt the following silicate melt solubility laws for $CO_2$, $H_2O$[89], and $H_2$[90]:

$$\ln\left(\frac{m_{CO_2} M_{melt}}{M_{CO_2}}\right) = \frac{m_{H_2O} M_{melt}}{M_{H_2O}} d_{H_2O} + a_{CO_2} \ln(fCO_2) + F_1 \quad (8)$$

$$\ln\left(\frac{m_{H_2O} M_{melt}}{M_{H_2O}}\right) = a_{H_2O} \ln(fH_2O) + F_2 \quad (9)$$

$$\log_{10}(m_{H_2} \times 10^6) = 0.524139 \times \log_{10}(fH_2) + 1.100836 \quad (10)$$

Here, $M_X$ are molar masses of respective species (kg/mol), and $M_{melt} = 0.06452$ kg/mol is the average molar mass of silicate melts. The terms $m_X$ represent the mass fraction of dissolved volatile species in the melt phase (kg X/kg melt), and the remaining terms are empirical constants, or weakly pressure-dependent constants in the case of $F_1$ and $F_2$.

We also permit graphite precipitation under C-rich, highly reducing conditions. Specifically, if the dissolved $CO_2$ mass fraction, $m_{CO_2}$, is less than graphite saturation then the melt is undersaturated and all carbon is partitioned between $CO_2$, CO, and $CH_4$. Conversely, if dissolved carbon exceeds graphite saturation then the excess C partitions into solid graphite (total moles $n_{graphite}$) and the remaining dissolved carbonate concentration in the melt equals graphite saturation (see supplementary materials for details).

The speciation equations above are constrained by mass conservation across all phases. Hydrogen atom conservation is given by:

$$n_H = n_{atm}\left(4\frac{fCH_4}{P} + 2\frac{fH_2O}{P} + 2\frac{fH_2}{P}\right) + 2m_{H_2O}\frac{\Pi_{melt}}{M_{H_2O}} + 2m_{H_2}\frac{\Pi_{melt}}{M_{H_2}} \quad (11)$$

Here, $n_H$ is the total number of moles of H, $n_{atm}$ is the total number of moles in the atmosphere, P is surface pressure, and $\Pi_{melt} = \rho_m V_m$ is the mass of the magma ocean (kg).

Carbon atom conservation is similarly given by:

$$n_C = n_{atm}\left(\frac{fCH_4}{P} + \frac{fCO_2}{P} + \frac{fCO}{P}\right) + m_{CO_2}\frac{\Pi_{melt}}{M_{CO_2}} + n_{graphite} \quad (12)$$

Here, $n_C$ is the total moles carbon in the fluid system. Note that we are ignoring CO and $CH_4$ dissolution because dissolved carbon dioxide and graphite are expected to dominate. This is a conservative assumption for atmospheric loss since adding further reduced species would potentially shield dissolved carbon from escape.

Finally, we impose oxygen atom conservation. Magma ocean models often neglect oxygen conservation on the grounds that the mantle is an infinite reservoir of oxygen atoms. This assumption is not justified for large surface volatile inventories where the reaction of molecular $H_2$ with the silicate interior can potentially modify bulk mantle redox; if oxygen conservation is not imposed, then more water can be created than there are oxygen atoms in the mantle. Oxygen conservation is given by the following equation:

$$n_O + n_{O-bound-Fe} = n_{atm}\left(2\frac{fH_2O}{P} + 2\frac{fCO_2}{P} + \frac{fCO}{P} + 2\frac{fO_2}{P}\right) + 2m_{CO_2}\frac{\Pi_{melt}}{M_{CO_2}} + m_{H_2O}\frac{\Pi_{melt}}{M_{H_2O}} \quad (13)$$

Here, $n_O$ is the total number of "free" oxygen atoms, that is, those not bound up as FeO, $SiO_2$, $Al_2O_3$ etc. which we do not explicitly consider. Free oxygen is partitioned between $FeO_{1.5}$ and atmospheric + dissolved volatile species. It is necessary to include a correction $n_{O-bound-Fe}$ for free oxygen bound up in ferric iron (0.5 mol per $FeO_{1.5}$)

and for instances where additional free oxygen is liberated by the reduction of ferrous iron to metallic iron i.e. $FeO + H_2 = H_2O + Fe$ (we return to the formalism for this below). Since the model conserves H, O, C, and Fe atoms, redox is conserved by construction.

Finally, total surface pressure and the total number of moles in the atmosphere are related as follows:

$$P = \frac{n_{atm}g}{4\pi r_p^2 \times 10^5}\left(M_{CH_4}\frac{fCH_4}{P} + M_{CO_2}\frac{fCO_2}{P} + M_{CO}\frac{fCO}{P}\right.$$
$$\left. + M_{H_2}\frac{fH_2}{P} + M_{H_2O}\frac{fH_2O}{P} + M_{O_2}\frac{fO_2}{P}\right) \tag{14}$$

Where the factor of $10^5$ is necessary to convert between fugacities (bar) and SI units of pressure (Pa). If oxygen fugacity is treated as a free variable, then our gases plus dissolved species code closely reproduces that in Gaillard, et al.[33].

Since mantle redox is co-evolving with the atmosphere, we need to relate oxygen fugacity to iron speciation in the melt. For oxidized regimes this is governed by the empirical relationship between ferrous and ferric iron in Kress and Carmichael[91]:

$$\ln\left(\frac{X_{FeO_{1.5}}}{X_{FeO}}\right) = 0.196 \times \ln(fO_2) + c(T,P,X_i) \tag{15}$$

Here, $X_{FeO_{1.5}}$ and $X_{FeO}$ are the molar fractions of ferric and ferrous iron in the melt, respectively, and $c(T,P,X_i)$ represents a series of T-dependent, P-dependent, and melt-composition dependent coefficients (see Supplementary Materials); metallic phases are assumed to be negligible under oxidizing conditions.

In contrast, for low oxygen fugacities, ferric iron is negligible and oxygen fugacity is governed by the relationship between metallic iron and ferrous iron in the melt:

$$2 \times \log_{10}\left(\frac{1.5X_{FeO}}{0.8X_{Fe}}\right) = \log_{10}(fO_2) - \log_{10}(fO_2(IW)) \tag{16}$$

Here, the oxygen fugacity at the iron wustite (IW) buffer is calculated from surface temperature and pressure using Frost[92]. The mole fraction of iron in the metallic phase, $X_{Fe}$ equals 1, and activity coefficients representative of typical surface conditions have been chosen for iron oxide, 1.5, and metallic iron, 0.8[93,94]. For greater numerical efficiency, we imposed a smooth transition in oxygen fugacity between equation (i) and (ii) at intermediate redox states, as described in the supplementary materials. In all cases, we enforce conservation of iron in the melt (although in some instances we permit total iron to evolve with time if iron is sequestered into the metallic core, as described below):

$$XFeO + XFeO_{1.5} = XFeO_{Tot} \tag{17}$$

$$XFeO_{Tot}|_{t=0} = XFeO_{Tot} + \frac{m_{Fe}M_{melt}}{M_{Fe}} \tag{18}$$

Taken as a whole, given total silicate melt mass ($\Pi_{melt}$), the number of hydrogen ($n_H$), carbon ($n_C$), oxygen atoms ($n_O$), and the iron content of melt, our model simultaneously solves for surface volatile speciation at a given surface temperature. The model then iteratively calculates OLR, ASR, and heat flux from the interior, $q_m$, to solve for radiative-convective equilibrium and geochemical equilibrium every timestep (using the previous timestep as a guess for computational efficiency). Consistent with other magma ocean models, we assume the dissolved volatile concentration of the melt at the surface is equal to the dissolved volatile content at depth given increasing volatile solubility with pressure and the comparatively low viscosity of the magma ocean favoring rapid chemical equilibration (although see Salvador and Samuel[95] for possible limitations to this assumption).

Similarly, while we recognize that iron speciation and mantle redox is pressure sensitive[96–98], we only compute iron speciation and oxygen fugacity at the surface of the magma ocean since only the surface melt interacts directly with the atmosphere.

## Stellar evolution and atmospheric escape

Standard parameterizations of bolometric luminosity[99] and XUV luminosity evolution are adopted for the sun[100] and TRAPPIST-1[101]. Atmospheric escape is either diffusion limited or XUV limited, depending on the stellar XUV flux and the composition of the upper atmosphere, which is defined here as everything above the tropopause. To simplify escape calculations, $H_2O$, $CO_2$, $CH_4$, $H_2$ are assumed to photodissociate such that the only upper atmosphere species are atomic H, atomic O, and CO[102]. If the upper atmosphere is H-poor, then escape will be diffusion-limited as H escapes via diffusion through an O and CO background; neither O nor CO can escape in the diffusion-limited regime. In contrast, in the XUV-limited regime, the hydrodynamic outflow of H may drag along O (and even CO) following previous parameterizations[103–105]. Thermosphere temperature and the cold-trap temperature modulate the drag of heavier species and the amount of H-bearing gases that reach the upper atmosphere, respectively. The escape model also incorporates broad parameterizations of XUV-driven escape efficiency, and a smooth transition from the diffusion-limited to XUV-limited regime. Our escape scheme is described in Supplementary Materials and is similar to that of Krissansen-Totton, et al.[28], except that we assume that carbon escapes as CO rather than $CO_2$ (a conservative assumption that makes the loss of C easier), and a correction is made for $CH_4$ in reducing atmospheres.

## Time evolution

Volatile reservoirs co-evolve with the solidification of the magma ocean as governed by the following system of equations. We distinguish between the solid reservoirs (silicate mantle below solidus), and fluid reservoirs (atmosphere + volatiles dissolved in the magma ocean).

The time evolution of solid, $\Pi_{H-solid}$ (kg), and fluid, $\Pi_{H-fluid}$ (kg), hydrogen reservoirs is governed by the following equations:

$$\frac{d\Pi_{H-solid}}{dt} = 4\pi\rho_m r_s^2 \frac{dr_s}{dt}\left[\left(m_{H_2O}\frac{M_{H_2}}{M_{H_2O}}\right)((1-f_{TL})k_{H2O} + f_{TL}) + m_{H_2}f_{TL}\right] \tag{19}$$

$$\frac{d\Pi_{H-fluid}}{dt} = -\frac{d\Pi_{H-solid}}{dt} + \phi_H \tag{20}$$

Here, $f_{TL}$ is the melt fraction trapped in the mantle as the solidification front moves towards the surface[42,106], which in turn depends on the rate of change in mantle temperature, $k_{H2O} = 0.01$ is the partition coefficient for water between fluid and crystalline phases, and $\phi_H$ is the escape flux of H. In the supplementary materials we investigate the sensitivity of our results to melt trapping assumptions and find it has a negligible effect.

For free oxygen, the time-evolution equations governing solid, $\Pi_{O-solid}$ (kg), and fluid, $\Pi_{O-fluid}$ (kg), reservoirs are:

$$\frac{d\Pi_{O-solid}}{dt} = 4\pi\rho_m r_s^2 \frac{dr_s}{dt}\left[m_{FeO_{1.5}}\frac{M_O}{M_{FeO_{1.5}}} + m_{H_2O}\frac{M_O}{M_{H_2O}}((1-f_{TL})k_{H2O} + f_{TL})\right.$$
$$\left. + m_{CO_2}\frac{M_O}{M_{CO_2}}((1-f_{TL})k_{CO_2} + f_{TL})\right] - \chi_{metal} \tag{21}$$

**Table 1 | free parameters in our model, the nominal parameter value for illustrative calculations (Figs. 2–4), and the full Monte Carlo range shown in later calculations (Fig. 5)**

| Parameter name | Variable | Nominal point estimate | Monte Carlo range | Explanation/Reference |
|---|---|---|---|---|
| Initial conditions | Initial carbon, C (kg) | $4 \times 10^{20}$ | $10^{20}$–$10^{21.5}$ | Approximate bulk silicate Earth carbon inventory[111-113] |
| | Initial free oxygen, O (kg) | $6 \times 10^{21}$ | $10^{21}$–$10^{22}$ | Approximately Earth-like (ensures post-solidification mantle redox around quartz-fayalite-magnetite buffer if no nebular atmosphere). |
| | Initial hydrogen, H (kg) | $4 \times 10^{20}$ | $10^{20}$–$10^{23}$ | Point estimate is approximate bulk silicate Earth inventory, range encompasses large primary atmosphere. |
| | Initial radionuclide U, Th, and K inventory (relative Earth) | 1.0 | 0.33–30.0 | Scalar multiplication of Earth's radionuclide inventories in Lebrun, et al.[27]. Allows for modest tidal heating. |
| | Initial mantle FeO (mole fraction) | 0.06 | 0.06 | Earth-like value assumed for nominal calculations, but supplementary material investigates broad range of FeO (0.02 - 0.2) |
| Stellar evolution and escape parameters | TRAPPIST-1 XUV saturation time, $t_{sat}$ | 3.14 | $3.14^{+2.22}_{-1.46}$ Gyr | XUV evolution parameters drawn randomly from joint distribution[101]. |
| | Post saturation phase XUV decay exponent, $\beta_{decay}$ | −1.17 | $-1.17^{+0.27}_{-0.28}$ | XUV evolution parameters drawn randomly from joint distribution[101]. |
| | Saturated $\log_{10}(F_{XUV}/F_{BOLOMETRIC})$ flux ratio | -3.03 | $-3.03^{+0.25}_{-0.23}$ | XUV evolution parameters drawn randomly from joint distribution[101]. |
| | Escape efficiency at low XUV flux, $\varepsilon_{low}$ | 0.2 | 0.01–0.3 | See escape section in Krissansen-Totton, et al.[28]. |
| | Transition parameter for diffusion limited to XUV-limited escape, $\lambda_{tra}$ | 1.0 | $10^{-6}$–$10^{1}$* | See escape section in Krissansen-Totton, et al.[28]. |
| | XUV energy that contributes to XUV escape above hydrodynamic threshold, $\zeta_{high}$ | 50% | 0–100% | See escape section in Krissansen-Totton, et al.[28]. |
| | Cold trap temperature variation, $\Delta T_{cold-trap}$ | 0 K | −30 to +30 K | Cold trap temperature, $T_{cold-trap}$, equals planetary skin temperature plus a fixed, uniformly sampled variation, $T_{cold-trap} = T_{eq}(1/2)^{0.25} + \Delta T_{cold-trap}$. Here, $T_{eq}$ is the planetary equilibrium temperature given assumed albedo. |
| | Thermosphere temperature, $T_{thermo}$ | 1000 K | 200–5000 K* | 114–116 |
| Interior parameter | Mantle viscosity coefficient | 10 Pa | $10^1$–$10^3$ Pa s* | Solid mantle kinematic viscosity, $\nu_{rock}$, (m²/s) is given by the following equation: $\nu_{rock} = V_{coef} 3.8 \times 10^7 \exp(\frac{350000}{8.314 T_p})/\rho_m$ Here $T_p$ is mantle potential temperature (K) and $\rho_m$ is mantle density (kg/m³). See Krissansen-Totton, et al.[28]. |
| Albedo | Bond albedo during magma ocean solidification | 0.2 | 0.0–0.2 | 85,86 |

*Denotes this variable was sampled uniformly in log space. All others (except stellar XUV parameters) sampled uniformly in linear space.

$$\frac{d\Pi_{O-fluid}}{dt} = -4\pi\rho_m r_s^2 \frac{dr_s}{dt}\left[m_{FeO_{1.5}}\frac{M_O}{M_{FeO_{1.5}}} + m_{H_2O}\frac{M_O}{M_{H_2O}}((1-f_{TL})k_{H2O}+f_{TL})\right.$$
$$\left. + m_{CO_2}\frac{M_O}{M_{CO_2}}((1-f_{TL})k_{CO_2}+f_{TL})\right] - \phi_O + \psi_{metal}$$

$$\text{(22)}$$

Here, $k_{CO_2}$ is the partition coefficient for $CO_2$ between fluid and crystalline phases, and $\phi_O$ is the escape flux of O. The metal corrections, $\psi_{metal}$ and $\chi_{metal}$ are described below.

The time-evolution equations governing carbon reservoirs are as follows:

$$\frac{d\Pi_{C-solid}}{dt} = 4\pi\rho_m r_s^2 \frac{dr_s}{dt}\left(m_{CO_2}\frac{M_C}{M_{CO_2}}\right)((1-f_{TL})k_{CO_2}+f_{TL}) \quad \text{(23)}$$

$$\frac{d\Pi_{C-fluid}}{dt} = -\frac{d\Pi_{C-solid}}{dt} - \phi_C \quad \text{(24)}$$

Where C reservoirs and escape flux are similarly denoted. Note that we neglect transfer of graphite to the solid interior due to its buoyancy in silicate melts.

Finally, we include three additional equations to track the evolution of solid ferrous ($\Pi_{FeO-solid}$), ferric ($\Pi_{FeO_{1.5}-solid}$), and metallic iron ($\Pi_{Fe-solid}$) in the solid mantle interior:

$$\frac{d\Pi_{FeO_{1.5}-solid}}{dt} = 4\pi\rho_m r_s^2 \frac{dr_s}{dt}m_{FeO_{1.5}} \quad \text{(25)}$$

$$\frac{d\Pi_{FeO-solid}}{dt} = 4\pi\rho_m r_s^2 \frac{dr_s}{dt}m_{FeO} \quad \text{(26)}$$

$$\frac{d\Pi_{Fe-solid}}{dt} = 4\pi\rho_m r_s^2 \frac{dr_s}{dt}m_{Fe} \quad \text{(27)}$$

We are assuming surface speciation of iron governs the portioning of iron into solid phases, which is an oversimplification of redox stratification. However, since we are most interested in speciation at the surface, and because the mantle is assumed not to interact with the atmosphere post-solidification (see below), this simplification is appropriate.

The time evolution of the solidification radius, $r_s$, is governed by the movement of the solidus-mantle adiabat intercept as the mantle cools, and is calculated analytically[28].

## Treatment of metallic iron
The system of equations described above are agnostic on the fate of metallic iron that forms in the melt. Rather than attempt to solve the complex coupled geodynamical + geochemical problem of core formation, we instead consider two endmember scenarios for metallic iron:

(1) All metallic iron remains entrained in the silicate melt, until partitioned into the solid silicate mantle via Eq. (27). This endmember is plausible given the persistence of metallic iron in Mercury's reducing crust[107] as well as plausible proposed mechanisms for iron droplet entrainment or buoyancy[108,109]. In this scenario, the metal correction terms are given by:

$$\chi_{metal} = \psi_{metal} = 4\pi\rho_m r_s^2 \frac{dr_s}{dt}m_{Fe}\frac{M_O}{M_{Fe}} \quad \text{(28)}$$

This ensures the transfer of metallic iron to the solid mantle permanently adds free oxygen to fluid phases. Simultaneously, we apply the following correction to fluid phase equilibria for free oxygen

liberated by metallic iron in the melt:

$$n_{O-as-Fe} = \Pi_{melt}\frac{M_O}{M_{Fe}}(XFeO_{Tot}|_{t=0} - XFeO_{Tot})\frac{M_{Fe}}{M_{melt}} - \frac{\Pi_{melt}m_{FeO_{1.5}}}{M_O}\left(\frac{0.5M_O}{M_{Fe}+1.5M_O}\right)$$

$$\text{(29)}$$

Here $XFeO_{Tot}|_{t=0}$ is a constant, but $XFeO_{Tot} = X_{FeO} + X_{FeO_{1.5}}$ may evolve with melt redox as iron is partitioned between oxidized and metallic phases.

(2) All metallic iron is instantaneously removed to the metallic core. This is plausible on the grounds that iron droplets have a much higher density than the surrounding silicate melt and are expected to sink on timescales shorter than the typical magma ocean solidification[110]. In this scenario,

$$\chi_{metal} = 0 \quad \text{(30)}$$

$$\psi_{metal} = 4\pi\rho_m r_s^2 \frac{dr_s}{dt}m_{Fe}\frac{M_O}{M_{Fe}}(XFeO_{Tot}|_{t=0} - XFeO_{Tot})\frac{M_{Fe}}{M_{melt}} \quad \text{(31)}$$

Here, we assume the solidification of the magma ocean does not add free oxygen-consuming metallic iron to the solid mantle because that metal instantly moves to the core. In contrast, free oxygen is still permanently liberated in the fluid phase when melt with a lower-than-initial FeO content solidifies. In this case, the free oxygen in the fluid phase is similarly corrected using Eq. (29).

## Post magma ocean evolution
Our focus is on the transition from primary to secondary atmosphere evolution, and so we do not attempt to model the full diversity of processes shaping atmospheric evolution after the surface temperature drops below the solidus. However, to allow comparisons between outcomes of different magma ocean duration, we do continue atmospheric evolution after magma ocean solidification permitting atmospheric escape to continue alongside stellar evolution, and continuous re-equilibration of surface volatiles at a plausible quench temperature (1000 K); surface climate evolves self-consistently with atmospheric composition and stellar evolution. No exchange of volatiles between the atmosphere and interior occurs post-magma ocean solidification. This is a conservative assumption for investigating atmospheric preservation since real planets may replenish escaping volatiles by magmatic degassing. The end of the magma ocean is defined as when the surface temperature drops below the solidus; volatiles dissolved in the magma ocean at this time are assumed to remain trapped in the mantle, thereby minimizing the surface volatile inventory susceptible to subsequent escape.

## Monte Carlo approach
To accommodate uncertainty in initial composition, atmospheric escape, atmosphere-interior interaction, and interior evolution processes, we present both nominal "point estimate" calculations, and results from Monte Carlo ensembles that broadly sample uncertain planetary parameters. Table 1 includes the unknown free-parameters in our model, the nominal parameter value for illustrative purposes, and the full Monte Carlo range shown in later calculations.

## Data availability
Full time-evolution outputs from nominal model calculations have been permanently archived on Zenodo: 10.5281/zenodo.13161895

## Code availability
The Python code for the PACMAN-P magma ocean evolutionary model is available on the lead author's Github and has also been permanently archived on Zenodo: 10.5281/zenodo.13206993

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

## Acknowledgements

J.K.T acknowledges support from NASA Astrophysics Precursor Science Grant 80NSSC23K1471 and the Virtual Planetary Laboratory, which is a member of the NASA Nexus for Exoplanet System Science, and funded via NASA Astrobiology Program Grant 80NSSC23K1398.

## Author contributions

J.K.-T. designed the study and performed calculations, N.W. provided numerical methods expertise and contributed to magma ocean code development, M.T. contributed to the climate model development, J.J.F. provided expertise on radiative convective climate models and sub-Neptune evolution. All authors contributed to drafting and editing the manuscript.

## Competing interests

The authors declare no competing interests.
