## [Peer Review File · Nature Communications]

REVIEWER COMMENTS

Reviewer #1 (Remarks to the Author):

General Comments

This paper presents a new coupled model for investigating the evolution of atmospheres around terrestrial planets, including atmospheric escape and ocean magma model, for a diverse and large array of possible compositions (both reduced and oxidized) and other parameters. The paper is easy to follow, with a good level of detail. Results and model limitations are well discussed, the methodology and the conclusions seem robust. Furthermore, implications for assessing habitability of extrasolar planets (based on the specific case of the TRAPPIST-1 system) are potentially important. I therefore recommend publication once the minor issues detailed below have been addressed.

Important caveat: I can only provide my expertise in the atmospheric modeling aspects of this manuscript. All other comments will be rather general, from a non-specialist point-of-view.

Specific Comments

* l.404-414: the importance of modeling the volatile cycle post-magma ocean stage is indeed of paramount importance, as the authors underline, and are probably the next step in further studies. Nevertheless, some quick checks may be performed at this stage in order to qualitatively assess which weathering processes are expected and better constrain possible habitability. For example, checking that the pressure/temperature at the bottom of water oceans is compatible or not with the formation of high-pressure dense water ice, which may impede hydrosphere/interior exchanges and dissolution of crust mineral salts in the oceans (probable prerequisite for habitability).

* l.491: hybrid continua (e.g. CO₂-H₂O, H₂-He, etc.) should be included in the opacity calculations, and will strengthen atmospheric opacity. Also, since helium is a major component of primary atmospheres, inclusion of H₂-He and He-He continua should be considered also (even though He alone is not a spectroscopically active species). Also, why did you assume that CO-CO continuum is

identical to N2-N2 (maybe for isoelectronicity reasons)?

* I.504: this approach with respect to albedo should be justified (e.g. high sensitivity of albedo to cloud microphysics, atmospheric dynamics and horizontal contrasts which would require 3D modeling beyond the scope of the study). Also, higher values should be considered in testing (see below).

* I.508-510: a suggestion to suppress some numerical instabilities would be to allow for a more detailed energy balance beyond balancing fluxes. Energy is stored/released by various processes (mostly phase changes), which each providing an additional heat flux (when divided by the simulation timestep) that needs to be accounted in the energy balance. A quick order-of-magnitude check of the possible fluxes could be performed in order to assess a possible stabilization (adding energy reservoirs would lead to a larger inertia and improved numerical stability).

* I.510-517: the authors are right to point out this effect. Its influence upon simulations results could already be assessed in the supplementary material, assuming e.g. a isothermal profile for $P > 100$ bar, below the convective zone (100 bar is conservative estimate of the minimal transition pressure, since Venus does not host such a deep radiative zone at this surface pressure).

* Table 1: if thick water clouds are present after magma ocean solidification, albedo may be significantly higher than 0.2 (as high as 0.6 around M-stars according to the already cited Pluriel et al. 2019 study) -- this would be likely if the moist adiabat spans a high and thick enough pressure range. Sensitivity check with respect to higher albedo values could be added in Supplementary Material.

* Sup, I.205-207: since escape takes place in the exosphere well above the homopause, mixing ratios of lighter (resp. heavier) species are enhanced (resp. lowered) at the exobase compared to their bulk atmospheric values. This "sedimentation" effect could be easily included, assuming each escaping species follows its own scale height above the homopause (the altitude difference between homopause and exobase should be estimated self-consistently). If this effect is already included in the computations, please make it more explicit.

Reviewer #1 (Remarks on code availability):

A README file is provided, with instructions about how to install and run the code. I did not attempt to run it myself, though.

Reviewer #2 (Remarks to the Author):

The manuscript by Krissansen-Totton and co-authors presents a new theoretical estimate for the transition from primary to secondary atmosphere, with a focus on the role of redox reactions and associated residual volatiles in the interior and atmosphere of rocky exoplanets, in particular TRAPPIST-1b and e.

The paper is well written and presents an extended, sophisticated modelling framework from the authors' previous work. The results are extensive, the methods are well documented and tested, and the results are highly relevant for the current debate surrounding the nature of secondary atmospheres. I have a few comments, but only one critical comment that is essential for publication (point 3), and I thus recommend publication after minor revision.

1) H accretion/loss versus volatile "ice" enrichment as cause of the radius valley + nature of primordial volatile inventory:

- The authors take a clear stance for H/He accretion+loss, and against enrichment by accretion of volatile-rich planetesimals or pebbles. Multiple authors, in particular Venturini et al. (2020) and Izidoro et al. (2022) have presented convincing models of the radius valley that align with modern accretion theories. On the scales you are probing here, water-ice envelopes for the outer TRAPPIST-1 planets are still very much a possibility (e.g., Agol et al. 2021, and many newer works), and therefore the range of free oxygen tested in the paper may not be fully representative for the possible initial conditions. I suggest either testing the possibility of up to ~20wt% enrichment in water ice (in addition to primordial H/He), or appropriately caveating the model range/scenario.
- Be that as it may, the language throughout the manuscript related to "chondritic" volatile abundances introduces a substantial confusion on the nature of heavier atmospheric species. What is "chondritic" for exoplanets? Even within the Solar System, the term "chondritic" carries so much historical baggage that it is not a quantitatively descriptive term. I suggest removing any reference to "chondritic" given the range of initial volatile inventory tested here. To give a motivation, the very recent work by Grewal et al. (2024) demonstrates (again) that the term "chondritic" is essentially meaningless: meteoritic bodies at present-day are highly evolved relative to the primordial nature and composition of the initial planetesimals and pebbles that formed the terrestrial planets; present-day chondritic meteorites span orders of magnitudes in H, C, N, S abundances due to planetary processing. Calling volatile abundances of extrasolar planets "chondritic" is not a useful term in this context. A reasonable summary of many of these processes are given in, for example, Krijt et al. (2023, PPVII).

2) Missing references:

The authors neglect a range of studies that have previously demonstrated several of the conclusions the authors arrive at. I do think that this paper here adds an important additional layer of complexity and applaud the authors for their effort in this. Nevertheless, they may want to appropriately attribute credit. In particular:

- Kimura & Ikoma (2020, 2022): This paper, while not including many of the physics and chemistry

included in this calculation here, already predicted many similar outcomes here, and is generally highly relevant for the type of model. I suggest giving proper credit to these works.

- Ikoma et al. (2012) for what is now called "core-powered mass loss" and cited as escape powered by internal heat flow.

- In the context of the discussion on page 13/14, I presume the recent string of K2-18b results (that appeared while this paper here was already in submission) becomes relevant. In particular the papers of Shorttle et al. (2024) and Wogan et al. (2024) explore connections between the deep atmospheric chemistry and the partitioning into the molten silicates in the reduced regime that is highly relevant for this paper here as well.

- Lines 462-465: A similarly/more relevant citation than Rubie et al. (2011) would be Chen & Jacobson (2022).

3) Key sentence in the conclusions:

"This potentially expands the range of habitable environments in the universe." Expanding relative to what? This sentence cannot be objectively derived from the results, as it implicitly states that the prevailing view had been that habitable zone M-dwarf exoplanets do not have volatile inventories at all. Many authors have stated this before, partly based on very similar reasoning as in this work, partly because of mechanism that are not treated here. A key uncertainty remains the planet formation path and thus the initial amount and fractionation of volatiles. It seems to me that only because the authors choose to not model ~wt% enrichment in water and other ices, as planet formation models suggest, complete removal of atmospheric volatiles is even a possibility in the modelling results. I suggest removing this sentence altogether.

Reviewer #3 (Remarks to the Author):

"The loss of primary atmospheres does not preclude habitability: A self-consistent model of the transition from sub-Neptune to terrestrial atmospheres"

This paper employs a coupled atmosphere-interior evolution model to estimate atmospheric changes during magma ocean solidification. It primarily focuses on predicting the habitability of TRAPPIST-1b and 1e; nevertheless, the model's applicability extends to terrestrial planets in general, offering a reliable framework for future studies. Prior works by the authors have already utilized a similar model, and I am convinced by the results in general. While reading the manuscript, however, a few questions came up, and I believe it would be helpful for the readers to address some of my inquiries within the manuscript.

If I have interpreted the results correctly, the key finding for TRAPPIST-1e is the formation of a H₂O-rich atmosphere primarily driven by H escape and the reduction of FeO. However, during the evolution, multiple processes appear to be at play — the retention of H₂O in the magma ocean, followed by the rapid expulsion of H₂O from the mantle during the final stages of solidification, and subsequently, the reaction between carbon monoxide (CO), hydrogen (H₂), among other species, at the time of surface solidification. Providing a little more details of each process, together with how transition occurs from a magma ocean to a solidified mantle would be helpful to understand the results.

Related to my previous comment, the final composition of the atmosphere appears to be the outcome of a complex interplay of various phenomena, including mantle degassing, atmospheric escape, and chemical reactions. The authors have demonstrated the robustness of their results through Monte Carlo analysis, but what exactly contributes to this robustness? For instance, could one envision an extreme scenario where XUV-driven escape to be 10 or 100 times higher than the nominal case, leading to the erosion of most of the hydrogen (H₂) before the solidification of a magma ocean?

In Figures 3 and 4, the vast majority (> 99.9%) of carbon (C) is degassed into the atmosphere throughout the evolution. This is in contrast to previous studies, such as those by Lebrun et al. (2013) and Salvador et al. (2017), which showed more than 10% of CO₂ in the magma ocean. While previous studies did not consider CO and CH₄, is the difference primarily attributed to differences in redox state, or is it more due to the adopted solubility law?

Line 652 — The description of f_{TL} in Krissansen-Totton and Fortney (2022) specifies that the factor f_{TL} is capped at 0.3. However, this may potentially underestimate the amount of volatiles trapped inside the mantle. In a vigorously convecting magma ocean, the bulk concentration remains constant, but volatile concentrations in the melt phase may vary as the fully molten surface and the partially molten state at the bottom of the magma ocean could have differing melt fractions. This non-uniform volatile concentrations may impact the overall trapping efficiency within the mantle.

Line 689 “... plausible proposed mechanisms for iron droplet entrainment (Lichtenberg, 2021)”^{[L][I][T][L]}_{[S][E][P][S][E][P]}— The authors suggest that whether metal is retained in the mantle or not has little influence on the overall results, so this may not affect the conclusion at all. However, I will point out that the retention of iron in the silicate magma, as proposed in Lichtenberg (2021), is only feasible when convection is vigorous. According to scaling by Solomatov (2015) and Lichtenberg (2021), grains smaller than 0.1 mm will settle for 10⁶ W/m², and 10 μm for 10⁴ W/m². Nevertheless, Figures 2-4 all depict a surface heat flux much lower than this value, suggesting that the entrainment of iron droplets seems unlikely.

Line 216-218 “we make the end member assumption that all metallic iron formed by the reduction of FeO is retained in the mantle, but results are qualitatively similar for the opposite case where all metallic iron is sequestered in the core (see below)”^{[L][I][T][L]}_{[S][E][P][S][E][P]}— The authors suggest that the limited influence of iron retention in the mantle on the overall results is mainly because of the small amount of FeO compared to the hydrogen (H) inventory from the primordial atmosphere. While exoplanets may indeed exhibit a wide range of FeO contents, ranging from none to a substantial amount of ferric iron, the authors do not explore this variability extensively. Is FeO content within the TRAPPIST system more likely to remain similar to that observed in the Solar System?^{[L][I][T][L]}_{[S][E][P][S][E][P]}

The rest is minor suggestions on the format etc.:

Figure 2

— Some lines are very hard to see, and I wasn't able to identify some of them (inc. H₂O (liquid) in Panel (e) and solid/molten Fe in Panel (f)). Since the main text indicates that no metallic Fe is produced in Figure 2, I suggest removing them from the label if they are not plotted.

Line 168-169 "The magma ocean ends after only a few $\sim 10^7$ years because there is insufficient water to maintain surface temperatures above the solidus."

— Is the weak greenhouse effect causing the rapid solidification? It would be helpful to explain the link between the small water amount and the short solidification timescale.

Line 555 "The terms m_X represent the melt fraction of dissolved volatile species"

— Some rephrasing will be helpful to improve the clarity: The "mass" fraction of dissolved volatiles "in the melt phase" ...?

Line 570, 581, ...

— I'm sorry to be picky, but I hope that a unified font style will be used in the published version (for example, CO_2 being italicized in Eq. (12) but H_2O being in regular font in Eq. (11)).

Reviewer #3 (Remarks on code availability):

The code is well documented, and it should be accessible to the readers.

We thank the reviewers for their constructive comments. The arguments in the manuscript are considerably more robust and persuasive because of this feedback. In response to reviewer comments, we have made the following major changes:

- New sensitivity tests have been added to explore the sensitivity of our results to albedo/clouds, melt trapping during magma ocean solidification, and isothermal vs. adiabatic deep atmospheres. We found that key results are generally insensitive to these assumptions.
- The results and discussion section have been substantially modified to more clearly explain competing effects driving atmosphere-interior evolution, and to describe the post magma ocean habitability implications more precisely/accurately.
- Our treatment of atmospheric escape is now explained in more detail with additional figures etc. in supplementary materials.

REVIEWER COMMENTS

Reviewer #1 (Remarks to the Author):

General Comments

This paper presents a new coupled model for investigating the evolution of atmospheres around terrestrial planets, including atmospheric escape and ocean magma model, for a diverse and large array of possible compositions (both reduced and oxidized) and other parameters. The paper is easy to follow, with a good level of detail. Results and model limitations are well discussed, the methodology and the conclusions seem robust. Furthermore, implications for assessing habitability of extrasolar planets (based on the specific case of the TRAPPIST-1 system) are potentially important. I therefore recommend publication once the minor issues detailed below have been addressed.

Important caveat: I can only provide my expertise in the atmospheric modeling aspects of this manuscript. All other comments will be rather general, from a non-specialist point-of-view.

Specific Comments

* I.404-414: the importance of modeling the volatile cycle post-magma ocean stage is indeed of paramount importance, as the authors underline, and are probably the next step in further studies. Nevertheless, some quick checks may be performed at this stage in order to qualitatively assess which weathering processes are expected and better constrain possible habitability. For example, checking that the pressure/temperature at the bottom of water oceans is compatible or not with the formation of

high-pressure dense water ice, which may impede hydrosphere/interior exchanges and dissolution of crust mineral salts in the oceans (probable prerequisite for habitability).

We have expanded a discussion-section paragraph on post-magma ocean evolution to include the quick checks suggested by the reviewer. To summarize: for the largest initial H₂ inventories we consider, the partial pressure of water at the surface of TRAPPIST-1e at the end of our calculations is several kbar (Fig. 5). Given the final surface temperature distribution for TRAPPIST-1e (Fig. S8) high pressure ices are not predicted to form at the base of the ocean. Of course, if TRAPPIST-1e accreted large amounts of water ice (a possibility highlighted by Reviewer 2) then larger surface water inventories are possible. But these quick checks show that the accretion of nebular gas and subsequent loss of the primary atmosphere does not, on its own, typically yield too much water to preclude habitability. A full exploration of habitability would require detailed studies of post-magma ocean evolution, since it is important to consider the operation of the deep water, the carbon cycle, and the connections with ocean chemistry and climate.

* I.491: hybrid continua (e.g. CO₂-H₂O, H₂-He, etc.) should be included in the opacity calculations, and will strengthen atmospheric opacity. Also, since helium is a major component of primary atmospheres, inclusion of H₂-He and He-He continua should be considered also (even though He alone is not a spectroscopically active species). Also, why did you assume that CO-CO continuum is identical to N₂-N₂ (maybe for isoelectronicity reasons)?

We believe our opacity calculations include the main sources of opacity for the atmospheres of interest, and that the errors introduced by any omissions are small compared to other model uncertainties. For example, we conducted full radiative-convective calculations with the Clima (Wogan et al., 2023) model for TRAPPIST-1e with and without H₂-He CIA. We considered the following atmospheric composition: p_{H₂} = 100 bar; p_{He} = 10 bar; p_{H₂O} = 10 bar; p_{CO₂} = 1 bar; p_{N₂} = 1 bar; p_{CO} = 1 bar; p_{CH₄} = 1 bar. The equilibrium surface temperature neglecting H₂-He opacity was 2720.9 K, and with H₂-He CIA included the surface temperature rises to only 2721.0 K.

To test the effect of H₂O-CO₂ CIA, we did another TRAPPIST-1e calculation with the following atmospheric composition: p_{CO₂} = 100 bar, p_{H₂O} = 37 bar (saturation), p_{N₂} = 10 bar. The resulting surface temperature without H₂O-CO₂ CIA is 520 K. When calculations are repeated using H₂O-CO₂ opacities (downloaded from here as HITRAN does not have H₂O-CO₂ CIA: https://web.lmd.jussieu.fr/~lmdz/planets/LMDZ.GENERIC/datagcm/continuum_data/H2O-CO2_continuum_MaTipping92.dat) then surface temperature increases to 528 K.

The reason these opacities are not that important is because there are lots of other strong absorbers in the atmosphere (e.g. H₂O-H₂O and H₂O line absorption, etc.) that dwarf any contribution from a minor CIA such as H₂-He or H₂O-CO₂. A summary of this discussion has been added to the methods section on radiative-convective climate model.

To our knowledge, CO-CO CIA data do not exist, and so, as the reviewer suggests, we picked N₂-N₂ as a probable analog based on isoelectronicity and molecular weight; this justification has been explained in the revised text.

* I.504: this approach with respect to albedo should be justified (e.g. high sensitivity of albedo to cloud microphysics, atmospheric dynamics and horizontal contrasts which would require 3D modeling beyond the scope of the study). Also, higher values should be considered in testing (see below).

We have added the following additional justification to the text explaining our approach to planetary albedo:

Attempting to model the complex cloud microphysics, aerosol properties, atmospheric circulation patterns, and climate system feedbacks that control planetary albedo across the primary-to-secondary atmospheric transition would be a challenging undertaking that would require a hierarchy of computational models including 3D GCMs. Instead, we take a conservative approach with respect to the duration of magma ocean solidification and assume a low albedo range (0-0.2). This range of values is consistent with what is expected for runaway greenhouse atmospheres (Kopparapu et al., 2013; Pluriel et al., 2019), but also maximizes the duration of the runaway greenhouse compared to cloudy scenarios, and by extension the longevity of hydrodynamic escape. We also now include sensitivity tests with higher albedo values (see response below).

* I.508-510: a suggestion to suppress some numerical instabilities would be to allow for a more detailed energy balance beyond balancing fluxes. Energy is stored/released by various processes (mostly phase changes), which each providing an additional heat flux (when divided by the simulation timestep) that needs to be accounted in the energy balance. A quick order-of-magnitude check of the possible fluxes could be performed in order to assess a possible stabilization (adding energy reservoirs would lead to a larger inertia and improved numerical stability).

We thank the reviewer for the suggestion; we will certainly consider this approach in future model applications. In this case, we are confident that the numerical instabilities are almost entirely attributable to the imperfections in our climate grid, since when the radiative-convective climate grid is replaced with a grey atmosphere and perfectly monotonic opacity-temperature relationships, the code speeds up significantly and numerical instabilities are mostly diminished.

* I.510-517: the authors are right to point out this effect. Its influence upon simulations results could already be assessed in the supplementary material, assuming e.g. a isothermal profile for $P > 100$ bar, below the convective zone (100 bar is conservative estimate of the minimal transition pressure, since Venus does not host such a deep radiative zone at this surface pressure).

We followed the reviewer's suggestion and recomputed a climate grid that assumes isothermal profiles below 100 bar. We then repeated nominal calculations for TRAPPIST-1e and presented these results as supplementary materials (Fig S12 and S13). Unsurprisingly, an isothermal deep

atmosphere leads to cooler surface temperatures and shorter duration magma oceans (Fig. S13). However, overall qualitative outcomes for TRAPPIST-1e are similar to the nominal model because the escape fluxes are most strongly influenced by luminosity evolution and specifically the runaway greenhouse duration, neither of which are changed by assuming a deep isothermal atmosphere. A new section with figures has been added to the supplementary materials to present these results, and they are also cited in the revised main text at the location noted by the reviewer.

* Table 1: if thick water clouds are present after magma ocean solidification, albedo may be significantly higher than 0.2 (as high as 0.6 around M-stars according to the already cited Pluriel et al. 2019 study) -- this would be likely if the moist adiabat spans a high and thick enough pressure range. Sensitivity check with respect to higher albedo values could be added in Supplementary Material.

We opted for a low albedo range (0-0.2) in the nominal to represent the cloud-free magma ocean state. Moreover, maintaining this low albedo throughout the planet's evolution is conservative with respect to atmospheric retention since higher albedos will result in cooler planets with shorter duration runaway greenhouse phases. However, to quantify the possible effect of clouds we conducted a sensitivity test as suggested by the reviewer that samples higher albedo values (0.5-0.7) representative of a continuously cloudy or hazy atmosphere. These results are explored in a new supplementary materials section (including Fig. S14 and S15). We find that higher albedos result in shorter magma ocean durations and slightly cooler surface temperatures, but otherwise do not dramatically affect results, except for resulting in slightly more secondary volatile retention.

* Sup, I.205-207: since escape takes place in the exosphere well above the homopause, mixing ratios of lighter (resp. heavier) species are enhanced (resp. lowered) at the exobase compared to their bulk atmospheric values. This "sedimentation" effect could be easily included, assuming each escaping species follows its own scale height above the homopause (the altitude difference between homopause and exobase should be estimated self-consistently). If this effect is already included in the computations, please make it more explicit.

We believe our Monte Carlo approach indirectly considers this "sedimentation" effect, as atmospheric molecules are converted to their atomic constituents and we sample a broad range in H abundances demarcating the transition from diffusion limited to XUV-limited escape, as discussed below. Explicitly including "sedimentation" in calculations would be non-trivial. This is because the difference between the exobase and homopause altitudes is temperature dependent, necessitating a model of upper atmosphere radiative transfer to self-consistently calculate the temperature profile.

We also note that our approach is implicitly pessimistic about secondary atmosphere retention since we are effectively assuming well-mixed (albeit atomized) constituents all the way up to the exobase; this maximizes the abundance of high molecular weight constituents vulnerable to XUV-driven hydrodynamic drag. Accounting for the sedimentation of heavier species would only lower the amount of C and O that could be dragged to space. Moreover, as discussed above the transition abundance parameter, λ_{tra} , is very broadly sampled; this implicitly accounts for

sedimentation and the uncertain bulk atmosphere H mixing ratio at which escape transitions from diffusion-limited to XUV-limited escape.

With all of that said, a more complete hydrodynamic escape calculation that self-consistently calculates upper atmosphere temperature and diffusive separation above the homopause would allow for more precise atmospheric loss predictions. A version of the discussion above along with equations showing how mixing ratios used for escape are calculated have been added to the supplementary materials.

Reviewer #1 (Remarks on code availability):

A README file is provided, with instructions about how to install and run the code. I did not attempt to run it myself, though.

Reviewer #2 (Remarks to the Author):

The manuscript by Krissansen-Totton and co-authors presents a new theoretical estimate for the transition from primary to secondary atmosphere, with a focus on the role of redox reactions and associated residual volatiles in the interior and atmosphere of rocky exoplanets, in particular TRAPPIST-1b and e.

The paper is well written and presents an extended, sophisticated modelling framework from the authors' previous work. The results are extensive, the methods are well documented and tested, and the results are highly relevant for the current debate surrounding the nature of secondary atmospheres. I have a few comments, but only one critical comment that is essential for publication (point 3), and I thus recommend publication after minor revision.

1) H accretion/loss versus volatile "ice" enrichment as cause of the radius valley + nature of primordial volatile inventory:

- The authors take a clear stance for H/He accretion+loss, and against enrichment by accretion of volatile-rich planetesimals or pebbles. Multiple authors, in particular Venturini et al. (2020) and Izidoro et al. (2022) have presented convincing models of the radius valley that align with modern accretion theories. On the scales you are probing here, water-ice envelopes for the outer TRAPPIST-1 planets are still very much a possibility (e.g., Agol et al. 2021, and many newer works), and therefore the range of free oxygen tested in the paper may not be fully representative for the possible initial conditions. I suggest either testing the possibility of up to ~20wt% enrichment in water ice (in addition to primordial H/He), or appropriately caveating the model range/scenario.

We thank the reviewer for highlighting these relevant papers on planet formation – they are now cited in the introduction where the radius valley is mentioned. We have also updated the description of our Monte Carlo calculations to explain that we have focused on H accretion to be most conservative with regards to atmospheric retention prospects – temperate terrestrial planets that accrete a large wt% of water will almost certainly retain surface volatiles, and this remains a possible explanation for the lower density of the outer TRAPPIST-1 planets as noted

by the reviewer. The revised discussion also now discusses the possibility of significant water accretion and cites the Agol et al. paper mentioned by the reviewer.

Our model is not built to accommodate extremely high water ice enrichment, so we are hesitant to include calculations with more oxidizing initial conditions. However, we did run a TRAPPIST-1b test case that we include here for illustrative purposes. Here, we increased initial H and O stoichiometrically to test the effect on atmospheric outcomes (we also consider a more restricted range of initial H masses). As expected, water-rich initial compositions are less likely to be completely desiccated, even for TRAPPIST-1b (the same is expected for TRAPPIST-1e).

- Be that as it may, the language throughout the manuscript related to "chondritic" volatile abundances introduces a substantial confusion on the nature of heavier atmospheric species. What is "chondritic" for exoplanets? Even within the Solar System, the term "chondritic" carries so much historical baggage that it is not a quantitatively descriptive term. I suggest removing any reference to "chondritic" given the range of initial volatile inventory tested here. To give a motivation, the very recent work by Grewal et al. (2024) demonstrates (again) that the term "chondritic" is essentially meaningless: meteoritic bodies at present-day are highly evolved relative to the primordial nature and composition of the initial planetesimals and pebbles that formed the terrestrial planets; present-day chondritic meteorites span orders of magnitudes in H, C, N, S abundances due to planetary processing. Calling volatile abundances of extrasolar

planets "chondritic" is not a useful term in this context. A reasonable summary of many of these processes are given in, for example, Krijt et al. (2023, PPVII).

We agree that the use of the term "chondritic" when referring to the heavier atmosphere inventories of exoplanets is not sufficiently quantitative, and it ignores the many ways in which primordial materials are processed to form terrestrial planets. We have therefore replaced the term "chondritic" with "BSE-like" whenever referring to exoplanet initial volatile inventories. This is appropriate since Bulk Silicate Earth has a specific quantitative meaning (as shown in Fig. 5), and because it is agnostic on the processes that give rise to terrestrial planet volatile inventories. Initializing terrestrial exoplanets with BSE-like volatile inventories is essentially exploring the hypothetical, "what if this exoplanet formed with the same initial volatiles as the Earth", a well-defined scenario that can be contrasted with model runs with large initial nebular atmospheres. The only place in the revised paper where we mention "chondritic" materials is when discussing solar system formation, specifically.

2) Missing references:

The authors neglect a range of studies that have previously demonstrated several of the conclusions the authors arrive at. I do think that this paper here adds an important additional layer of complexity and applaud the authors for their effort in this. Nevertheless, they may want to appropriately attribute credit. In particular:

- Kimura & Ikoma (2020, 2022): This paper, while not including many of the physics and chemistry included in this calculation here, already predicted many similar outcomes here, and is generally highly relevant for the type of model. I suggest giving proper credit to these works.
- Ikoma et al. (2012) for what is now called "core-powered mass loss" and cited as escape powered by internal heat flow.
- In the context of the discussion on page 13/14, I presume the recent string of K2-18b results (that appeared while this paper here was already in submission) becomes relevant. In particular the papers of Shorttle et al. (2024) and Wogan et al. (2024) explore connections between the deep atmospheric chemistry and the partitioning into the molten silicates in the reduced regime that is highly relevant for this paper here as well.
- Lines 462-465: A similarly/more relevant citation than Rubie et al. (2011) would be Chen & Jacobson (2022).

We thank the reviewer for bringing these highly relevant papers to our attention. We have incorporated all the previous studies mentioned above as follows:

- Kimura and Ikoma are now cited in both the results and the discussion section as examples of previous papers that predicting water-rich atmospheres from the reaction of nebular atmospheres with silicates on terrestrial planets around M-dwarfs.
- Ikoma and Hori (2012) has been added to the list of papers cited when discussing how "extreme ultraviolet (XUV) driven atmospheric escape and mass loss driven by interior heat flow both offer plausible [for atmospheric erosion from sub Neptunes]"
- Citations to Shorttle et al. (2014) and Wogan et al. (2024) have been added to that discussion. We also cite the recent TOI-270d observations (Benneke et al., 2024; Holmberg & Madhusudhan, 2024) which reveal C-rich atmospheres, as evidence against near-complete partitioning of C into metallic phases.

- Chen & Jacobsen (2022) has been added as another example of an N-body accretion + volatile partitioning model.

3) Key sentence in the conclusions:

"This potentially expands the range of habitable environments in the universe." Expanding relative to what? This sentence cannot be objectively derived from the results, as it implicitly states that the prevailing view had been that habitable zone M-dwarf exoplanets do not have volatile inventories at all. Many authors have stated this before, partly based on very similar reasoning as in this work, partly because of mechanism that are not treated here. A key uncertainty remains the planet formation path and thus the initial amount and fractionation of volatiles. It seems to me that only because the authors choose to not model ~wt% enrichment in water and other ices, as planet formation models suggest, complete removal of atmospheric volatiles is even a possibility in the modelling results. I suggest removing this sentence altogether.

We agree the original word was somewhat ambiguous and too sweeping. We have removed this sentence and rewritten the first conclusion as follows "For habitable zone planets, the transition from sub-Neptune to terrestrial planet via XUV-driven escape of a primary atmosphere typically does not strip the planet of high molecular weight volatiles, and instead is likely to leave behind large surface water inventories. This improves the habitability prospects for terrestrial plants around late M-dwarfs that accrete primary atmospheres."

Reviewer #3 (Remarks to the Author):

"The loss of primary atmospheres does not preclude habitability: A self-consistent model of the transition from sub-Neptune to terrestrial atmospheres"

This paper employs a coupled atmosphere-interior evolution model to estimate atmospheric changes during magma ocean solidification. It primarily focuses on predicting the habitability of TRAPPIST-1b and 1e; nevertheless, the model's applicability extends to terrestrial planets in general, offering a reliable framework for future studies. Prior works by the authors have already utilized a similar model, and I am convinced by the results in general. While reading the manuscript, however, a few questions came up, and I believe it would be helpful for the readers to address some of my inquiries within the manuscript.

If I have interpreted the results correctly, the key finding for TRAPPIST-1e is the formation of a H₂O-rich atmosphere primarily driven by H escape and the reduction of FeO. However, during the evolution, multiple processes appear to be at play — the retention of H₂O in the magma ocean, followed by the rapid expulsion of H₂O from the mantle during the final stages of solidification, and subsequently, the reaction between carbon monoxide (CO), hydrogen (H₂), among other species, at the time of surface solidification. Providing a little more details of each process, together with how transition occurs from a magma ocean to a solidified mantle would be helpful to understand the results.

We have expanded our explanations of Fig. 2, 3, and 4 to address this issue. The revised text explains why atmospheric composition is changing at different stages of the evolution (pre- and post-magma evolution), and tries to highlight which processes are dominating i.e. endogenous

water production, exsolution, oxidation via H₂-loss etc. For Fig. 3, in particular, we now better describe the full sequence of events from magma ocean solidification, surface water condensation, and the transition to an oxidized atmosphere due to continued H escape.

Related to my previous comment, the final composition of the atmosphere appears to be the outcome of a complex interplay of various phenomena, including mantle degassing, atmospheric escape, and chemical reactions. The authors have demonstrated the robustness of their results through Monte Carlo analysis, but what exactly contributes to this robustness? For instance, could one envision an extreme scenario where XUV-driven escape to be 10 or 100 times higher than the nominal case, leading to the erosion of most of the hydrogen (H₂) before the solidification of a magma ocean?

The Monte Carlo analysis encompasses a very broad range of possible escape fluxes. This is explained in the expanded “Escape parameterization” section which describes the Monte Carlo parameters controlling the efficiency of hydrodynamic escape, the transition from XUV-limited to diffusion-limited escape, and the extent to which XUV radiation above the crossover mass threshold drives additional hydrodynamic drag or is re-radiated. The Monte Carlo approach section in the main text also explains how a broad range of stellar XUV evolutions are folded into our calculations, following Birky et al. To more clearly illustrate the broad range of scenarios encompassed in our Monte Carlo analysis, we have also added supplementary Fig. S5, which shows the TRAPPIST-1 XUV luminosity evolution envelope, along with evolving escape fluxes for two atmospheric compositions for TRAPPIST-1b. The broad spread in escape fluxes, as well as the contrast between XUV-limited and diffusion-limited regimes directly addresses the robustness of the Monte Carlo analysis.

In Figures 3 and 4, the vast majority (> 99.9%) of carbon (C) is degassed into the atmosphere throughout the evolution. This is in contrast to previous studies, such as those by Lebrun et al. (2013) and Salvador et al. (2017), which showed more than 10% of CO₂ in the magma ocean. While previous studies did not consider CO and CH₄, is the difference primarily attributed to differences in redox state, or is it more due to the adopted solubility law?

Understanding exactly why carbon retention results differ between this paper and previous studies would require a careful model intercomparison. But we note that for our Earth-analog and Venus-analog calculations in the supplementary materials (Fig. S1 and S2, respectively), closer to ~2-10% of CO₂ is retained in the magma ocean, broadly in line with Lebrun et al. and Salvador et al. The Trappist-1e results with BSE volatile endowments are also similar (Fig. 2). We infer that much of the difference between Fig. 3+4 and previous studies is attributable to the reducing conditions created by the nebular atmosphere. Under reducing conditions, CO₂ is a small fraction of the total atmospheric carbon, and so the resultant dissolved carbonate melt fraction is similarly low. As noted in the paper, we neglect CH₄ and CO solubility – a conservative assumption that prevents reduced forms of C dissolving in the magma ocean, potentially shielding it from escape. Graphite solubility is included in our model, but as discussed in the supplementary materials, the reducing conditions created by the nebular atmosphere cause most C to partition into atmospheric CH₄, and so the resulting melt is undersaturated with respect to graphite due to low total dissolved carbon. A version of this discussion has been added to the “Earth and Venus Validation” section in supplementary materials.

Line 652 — The description of f_{TL} in Krissansen-Totton and Fortney (2022) specifies that the factor f_{TL} is capped at 0.3. However, this may potentially underestimate the amount of volatiles trapped inside the mantle. In a vigorously convecting magma ocean, the bulk concentration remains constant, but volatile concentrations in the melt phase may vary as the fully molten surface and the partially molten state at the bottom of the magma ocean could have differing melt fractions. This non-uniform volatile concentrations may impact the overall trapping efficiency within the mantle.

This is an interesting suggestion – we repeated nominal calculations with a maximum trapped melt fraction of 0.8. This is broadly representative of the upper limit to the melt-dominated creep to solid-state rheology (Costa et al. 2009, G3). Note that the melt fraction is calculated dynamically in the code as described in the cited literature, but increasing the maximum melt fraction trapped does allow for volatiles to be maximally retained in the mantle when solidification rates are high. We find that our conclusions are largely insensitive to melt trapping assumptions because solidification timescales are typically much longer than compaction timescales.

Line 689 “... plausible proposed mechanisms for iron droplet entrainment (Lichtenberg, 2021)” — The authors suggest that whether metal is retained in the mantle or not has little influence on the overall results, so this may not affect the conclusion at all. However, I will point out that the retention of iron in the silicate magma, as proposed in Lichtenberg (2021), is only feasible when convection is vigorous. According to scaling by Solomatov (2015) and Lichtenberg (2021), grains smaller than 0.1 mm will settle for 10^6 W/m^2 , and 10 μm for 10^4 W/m^2 . Nevertheless, Figures 2-4 all depict a surface heat flux much lower than this value, suggesting that the entrainment of iron droplets seems unlikely.

We acknowledge that entrainment seems unlikely for the heat fluxes we calculate. However, we believe the most conservative approach is to retain both endmember cases (i.e. all metal->core and all metal->mantle) since there may be other mechanisms by which metallic phases are retained in the silicate mantle. Rogers et al. (2024) suggests that droplets may become buoyant due to volatiles preferentially partitioning into metallic phases (see Discussion). A citation to Rogers et al. has been added, but we retain the two endmember cases to be conservative. It is also worth reiterating that overall conclusions are robust to assumptions about the fate of metallic iron.

Line 216-218 “we make the end member assumption that all metallic iron formed by the reduction of FeO is retained in the mantle, but results are qualitatively similar for the opposite case where all metallic iron is sequestered in the core (see below)” — The authors suggest that the limited influence of iron retention in the mantle on the overall results is mainly because of the small amount of FeO compared to the hydrogen (H) inventory from the primordial atmosphere. While exoplanets may indeed exhibit a wide range of FeO contents, ranging from none to a substantial amount of ferric iron, the authors do not explore this variability extensively. Is FeO content within the TRAPPIST system more likely to remain similar to that observed in the Solar System?

The “Sensitivity to initial mantle FeO” section in the supplementary materials tests the sensitivity of our results to an order of magnitude range of initial FeO contents (2-20%). As the reviewer notes, this encompasses the variability in silicate mantle material across solar system bodies. Moreover, Schlichting and Young (2022) consider equilibrium reactions between volatile, silicate, and metallic phases and report a relatively limited range of mantle FeO content (5-10%) across a wide range of H₂ envelopes, consistent with the sensitivity tests in this study. Stellar Fe abundances also tend to track those of other rock-forming elements (e.g. Mg, Si, Ca), suggesting planet-forming materials are not highly variable (Hinkel et al., 2014, AJ). While extremely iron-poor (or iron-rich) compositions cannot be ruled out, they seem unlikely given plausible cosmochemical abundances and expected equilibrium partitioning during accretion and differentiation. A summary of this discussion has been added to the FeO sensitivity test section in the supplementary materials.

The rest is minor suggestions on the format etc.:

Figure 2

— Some lines are very hard to see, and I wasn’t able to identify some of them (inc. H₂O (liquid) in Panel (e) and solid/molten Fe in Panel (f)). Since the main text indicates that no metallic Fe is produced in Figure 2, I suggest removing them from the label if they are not plotted.

Liquid H₂O and solid/molten Fe have been removed from the legends of Fig. 2 as suggested.

Line 168-169 “The magma ocean ends after only a few $\sim 10^7$ years because there is insufficient water to maintain surface temperatures above the solidus.”

— Is the weak greenhouse effect causing the rapid solidification? It would be helpful to explain the link between the small water amount and the short solidification timescale.

Yes, the differences in solidification time are attributable to the differences in greenhouse warming (the evolution of absorbed stellar radiation is unchanged between Fig. 2 and 3). To clarify the role of greenhouse warming in magma ocean solidification time, several lines of explanation been added to the Results section text explaining Fig. 2 and 3. Note also that Fig. S8 shows solidification time as a function of initial H.

Line 555 “The terms m_X represent the melt fraction of dissolved volatile species”

— Some rephrasing will be helpful to improve the clarity: The “mass” fraction of dissolved volatiles “in the melt phase”...?

This has been reworded as suggested to improve clarity.

Line 570, 581, ...

— I’m sorry to be picky, but I hope that a unified font style will be used in the published version (for example, CO₂ being italicized in Eq. (12) but H₂O being in regular font in Eq. (11)).

We thank the reviewer for catching this. We have attempted to standardize italicization in the manuscript and will make sure to double check the final version for unified style.

Reviewer #3 (Remarks on code availability):

The code is well documented, and it should be accessible to the readers.

Other changes:

- We added a few lines on recent thermal escape literature to the discussion section.
- Nominal calculations were repeated after discovering a minor bug in the code (results were unchanged by this correction).
- The Monte Carlo range for the abundance transition parameter was adjusted slightly, and this also did not substantially change conclusions.

REVIEWERS' COMMENTS

Reviewer #1 (Remarks to the Author):

The authors have answered my previous remarks in a very meaningful way, and I recommend publication of the paper in its present state as soon as possible.

Reviewer #3 (Remarks to the Author):

The authors have addressed all my comments in the updated manuscript, and I believe the manuscript is now suitable for publication in Nature Communications.